# Midbrain node for context-specific vocalisation in fish

Eric R. Schuppe[1,3], Irene Ballagh[1,4], Najva Akbari[2,5], Wenxuan Fang[1,6], Jonathan T. Perelmuter[1], Caleb H. Radtke[1], Margaret A. Marchaterre[1] & Andrew H. Bass [1] ✉

Vocalizations communicate information indicative of behavioural state across divergent social contexts. Yet, how brain regions actively pattern the acoustic features of context-specific vocal signals remains largely unexplored. The midbrain periaqueductal gray (PAG) is a major site for initiating vocalization among mammals, including primates. We show that PAG neurons in a highly vocal fish species (*Porichthys notatus*) are activated in distinct patterns during agonistic versus courtship calling by males, with few co-activated during a non-vocal behaviour, foraging. Pharmacological manipulations within vocally active PAG, but not hindbrain, sites evoke vocal network output to sonic muscles matching the temporal features of courtship and agonistic calls, showing that a balance of inhibitory and excitatory dynamics is likely necessary for patterning different call types. Collectively, these findings support the hypothesis that vocal species of fish and mammals share functionally comparable PAG nodes that in some species can influence the acoustic structure of social context-specific vocal signals.

All classes of motor actions depend on the brain for patterning behaviour-specific muscle activity. A prominent example is vertebrate vocalisation, where the ability to flexibly generate temporally precise variations in muscle activity translates to distinct sounds or call types among both tetrapods and fishes (Fig. 1).Variations in call type appropriate to different socio-behavioural contexts are typically generated by the same set of vocal muscles driven by different temporal patterns of activity, which raises the question of how a complex motor network can both initiate and pattern acoustically distinct calls in different social contexts.

Recent investigations show that sound production is more widespread and ancient among non-mammalian vertebrates than previously thought[1–3]. By comparison, our knowledge of underlying neural mechanisms remains quite limited. Studies of mammals, including primates, map two main streams of descending vocal circuitry

pathways, corticobulbar and limbic–midbrain–bulbar, which are often considered to orchestrate the production of verbal and non-verbal vocal behaviours, respectively[4–10]. A corticobulbar pathway stream links motor cortex directly to hindbrain interneurons and/or motoneurons. The limbic stream includes forebrain regions that can influence these same downstream pools indirectly via the midbrain periaqueductal gray (PAG). In mammals, including humans, PAG lesions cause mutism, thus indicating an essential role of this brain area in vocalisation[10]. Two non-mammalian vocal clades, birds (Aves) and toadfishes (Osteichthyes) (Fig. 1), share similar organisational patterns. Songbirds, the subjects for most studies of non-mammalian vertebrate vocalisation, have direct and indirect telencephalic pathways to the hindbrain[11]. The former has a well-established role in song patterning[12], while the latter has a relay through the PAG that is relatively unexplored, but is a site for eliciting calls[13,14]. Toadfishes, a family

[1]Department of Neurobiology and Behavior, Cornell University, Ithaca, NY 14853, USA. [2]Department of Applied and Engineering Physics, Cornell University, Ithaca, NY 14853, USA. [3]Present address: Department of Physiology, University of California San Francisco School of Medicine, San Francisco, CA 94305, USA. [4]Present address: Department of Zoology, The University of British Columbia, Vancouver V6T 1Z4 BC, Canada. [5]Present address: Department of Biology, Stanford University, Palo Alto, CA 94305, USA. [6]Present address: Graduate Program in Neuroscience, The University of British Columbia, Vancouver V6T 1Z4 BC, Canada. ✉e-mail: ahb3@cornell.edu

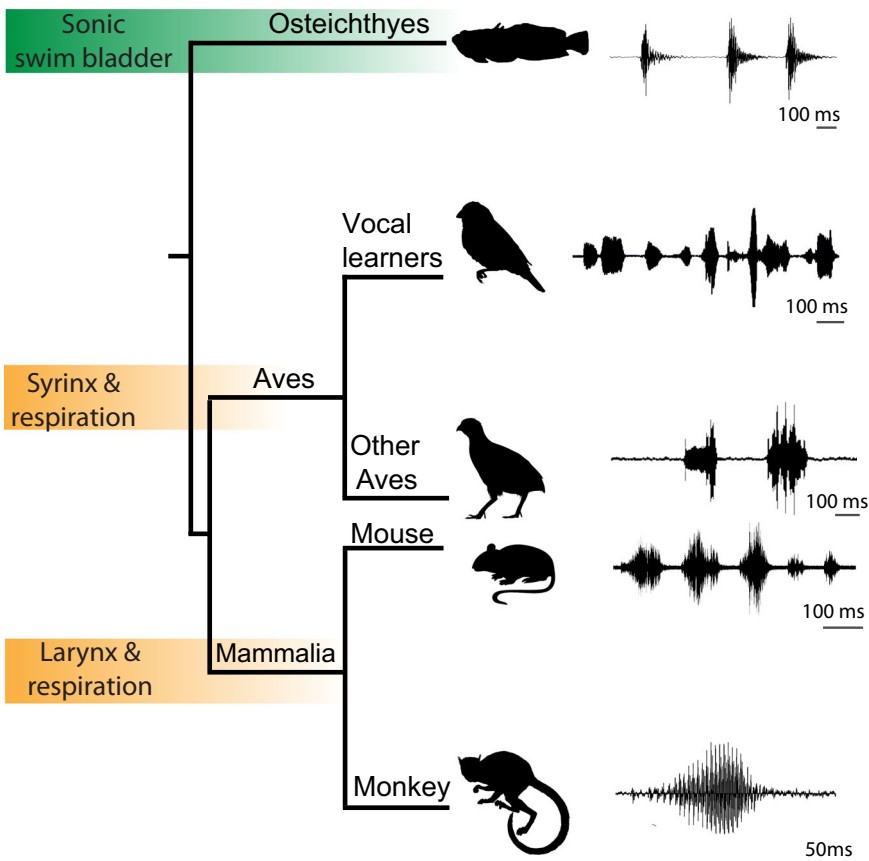

**Fig. 1 | Vocal diversity in fishes and tetrapods.** Despite the diversity of vocal organs among vertebrate lineages, e.g., sonic swim bladder, syrinx and larynx, the available evidence suggests shared patterns of vocal-acoustic features. Shown here are oscillogram traces of plainfin midshipman fish grunt train (*Porichthys notatus*, focal species of current study), estrildid finch song (*Lonchura striata domestica*) song, Japanese quail call (*Coturnix japonica*, other Aves; from xeno-canto [XC266707]; recordist: Albert Lastukhin), house mouse ultrasonic vocalisations (*Mus musculus;* from DeepSqueak[72]), and squirrel monkey caw (*Saimiri sciureus*). Animal sillouttes are from PhyloPic (creative commons).

of teleost fish (Batrachoididae), have a limbic stream resembling that of mammals, namely direct preoptic-anterior hypothalamic input to a midbrain PAG region that projects, in turn, directly to hindbrain vocal neurons[15]. Like mammals[10], the vocally active PAG region receives neuromodulator inputs[16–22]. Inactivation of this region with lidocaine or dopamine effectively silences forebrain-evoked vocal output[23,24]. Together, these studies suggest a critical role for a midbrain PAG region in vocal motor control between mammalian and non-mammalian vertebrate clades.

Although vocal regions within mammalian PAG are well documented (e.g.[25,26]), it remains unclear if and how individual PAG neurons, nodes and networks (see[27]) might enable the temporal patterning of context-specific vocalisations. Recognising this, we investigated PAG function in the plainfin midshipman fish (*Porichthys notatus*), a toadfish that generates temporally and spectrally distinct vocalisations specific to either courtship or aggression[28–31]. Males attract females to their nest at night during the breeding season by repetitively broadcasting long duration (up to 2 h each), multi-harmonic "hums" (Fig. 2a and Movie S1)[30,31]. When defending nests against intruders, males make "grunt trains", which are bouts of up to ~200 brief, broadband grunts (hundreds of msec each), and longer (msec-secs) "growls" with both broadband and multi-harmonic features similar to grunts and hums, respectively (Fig. 2b, c)[28–30]. Single grunts (Fig. 2b, far right), produced by both sexes, are akin to single notes, elements, or single note syllables in songbirds[32]. This simple repertoire of vocalisations, with each distinguished by social context and temporal features (duration, pulse repetition rate–PRR, amplitude–AMP), together with sound production machinery and vocal circuitry (Fig. 2d) that is

relatively simple and readily accessible for neurophysiological studies compared to tetrapods[33], make midshipman and other toadfishes ideal species to investigate a midbrain role in both initiating and patterning call types.

Here, we first measure immediate early gene (*c-fos*) expression to show that neurons in the caudolateral PAG of midshipman males (Fig. 2e, f) are active during courtship and agonistic calling. Using cellular compartment analysis of temporal activation of neurons by Fluorescent In Situ Hybridization (catFISH), we then show that individual PAG neurons are active during both courtship and agonistic or only agonistic calling, with few co-activated during a non-vocal behaviour, foraging. Uniform manifold approximation and projection (UMAP) analyses further demonstrate overlap between the temporal features of natural calls and "fictive calls" (vocal motor volley to sonic muscles) evoked by glutamate (GLU) microinjections into the PAG, but not its target in hindbrain vocal circuitry. In aggregate, the results provide the strongest evidence to date that the PAG can play an active role in both initiating and affecting the acoustic structure of social context-specific vocalisations. The results also support the hypothesis that fish and mammals have evolved a vocal PAG node that shares multiple functional attributes.

## Results
### Differential activation of PAG neurons during vocalisation
Prior studies of fish have not identified if, like mice[6–8], there is a specific PAG region that exhibits vocal-related neural activity dependent on behavioural state, e.g., courtship versus aggression. To investigate this question, we first quantified *c-fos+* expressing cells in nesting males

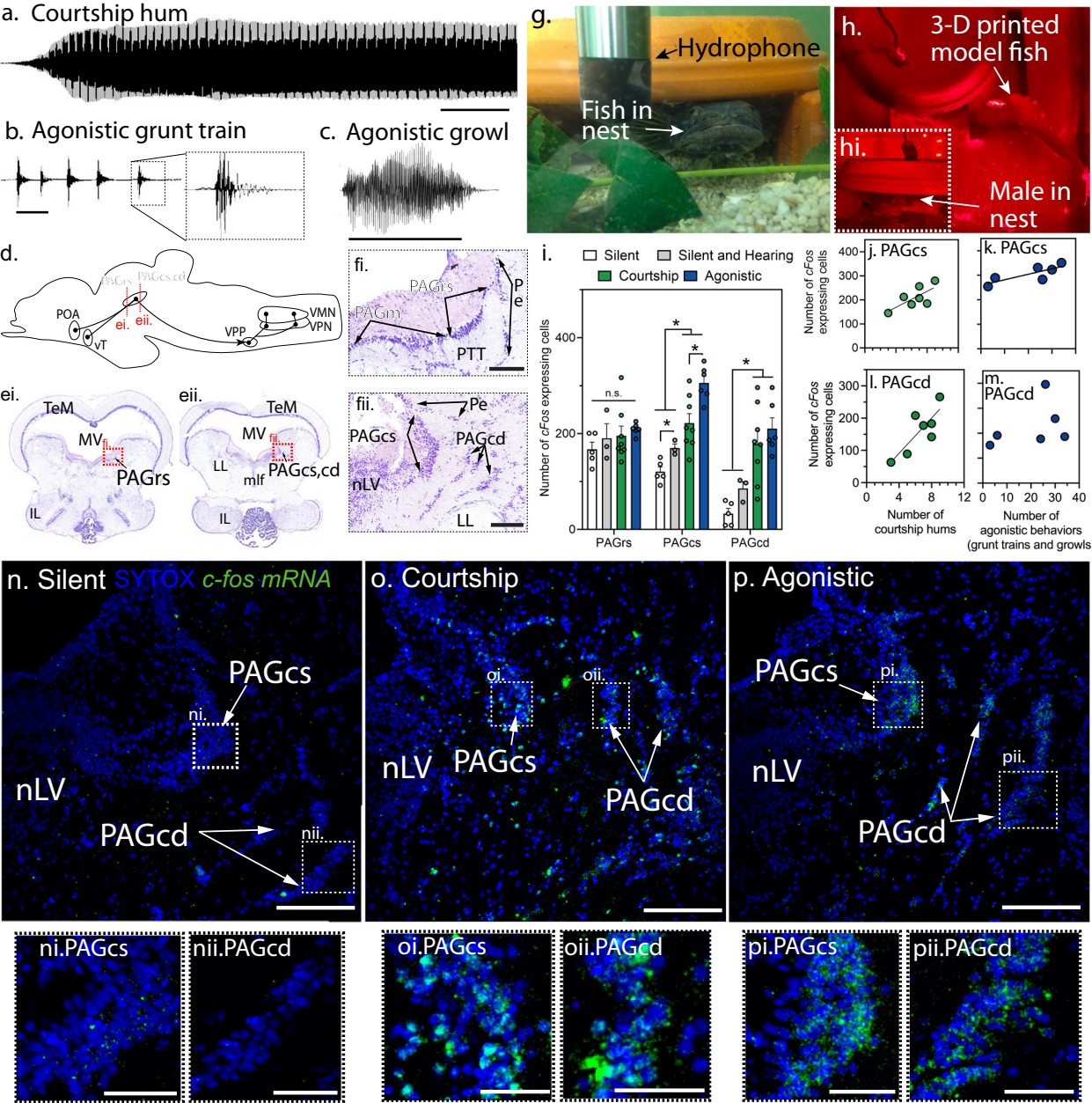

**Fig. 2 | Courtship and agonistic calling differentially activates midbrain periaqueductal gray (PAG). a–c** Male midshipman (*Porichthys notatus*) produce three, acoustically distinct vocalisations during the breeding season. Scale bars = 500 ms for **a–c**. **d** Midshipman vocal motor network, sagittal view. Abbreviations: PAG rostral (r) and caudal (c) zones; preoptic area, POA; vocal motor (VMN), vocal pacemaker (VPN) and vocal prepacemaker (VPP) nuclei ventral tuberal hypothalamus, vT. **ei**, **eii** Nissl-stained, coronal sections illustrating PAG zones (rostral and caudal superficial: rs, cs; caudal deep; cd; levels indicated in **d**); **fi**, **fii** are higher magnification views of **ei** and **eii**, respectively. Scale bars = 250 µm for **fi**, **fii**. LL lateral lemniscus, MLF medial longitudinal fasciculus, nLV nucleus lateralis valvulae, PAGm medial PAG, TeM midbrain tectum, Pe periventricular layer of auditory division of torus semicircularis, PTT paratoral tegmentum, PTT. **g** Male inside an artificial nest; photograph taken during lights-on to maximise clarity of the housing conditions. **h** Nesting male presented with 3-D printed model of midshipman male, picture taken at "night" under red light conditions. **hi** Photograph of male in the nest taken with a front facing camera. **i** Bar plots (mean ± SEM) illustrating number of *c-fos* mRNA expressing cells in PAGrs (ANOVA: *p* = 0.24; see table S2), PAGcs (ANOVA: *p* = 1.579e-05) and PAGcd (ANOVA: *p* = 0.0001) in silent (*n* = 5 animals), silent and hearing (*n* = 3), courtship calling (*n* = 8) and agonistic calling (*n* = 6) males. *denotes significant BH corrected *post-hoc* differences for Analysis of Variance (ANOVA) tests, *p* < 0.05. **j–m** Relationship of call number to *c-fos* expressing cell number in PAGcs and PAGcd. Solid trend lines denote significant relationships *p* < 0.05 (see table S2 for statistics ). **n–p** Representative staining of *c-fos* mRNA (green) in PAGcs and PAGcd (see Fig. S3 for PAGrs). Scale bars = 250 µm. **ni**, ii, **oi**, ii, **pi**, ii Scale bars = 50 µm.

isolated in separate aquaria who were silent (controls), spontaneously hummed[20,31], or made agonistic calls when presented with a 3D-printed model of a male at their nest entrance (Fig. 2g, h; "Methods" section; Movie S2). An initial survey revealed a striking elevation of *c-fos* label in a lateral periventricular midbrain region that we initially compared to vocally active regions of mammalian PAG (on the basis of brain stimulation and with tract-tracing)[15], and supported by subsequent studies (see [20,23,24,34,35]). This region is lateral to a markedly less cell dense medial periventricular region at rostral levels (PAGm) and the nucleus lateralis valvulae (nLV) at caudal levels (Fig. 2fi, fii) and medial throughout its extent to a thin neuronal layer (Pe; Fig. 2fi, fii) within the auditory division of the torus semicircularis, proposed homologue of

the inferior colliculus[36]. Elevated c-fos label was also apparent over 2-3 compact cell layers immediately ventrolateral to the caudal aspect of lateral periventricular region. We designated the latter a caudal deep zone—PAGcd, and the periventricular region aligned with it a caudal superficial zone—PAGcs; the rostral portion of the lateral periventricular region was designated the PAGrs (Fig. 2e, f). Both caudal zones are at the same level as the nLV. We previously showed PAG input to the hindbrain's vocal prepacemaker nucleus (VPP; formerly the ventral medullary nucleus, VM[37]) following neurobiotin injection into the caudal[15] or rostral[23] PAG. The latter injections were too large to distinguish separate caudal zones. However, a neurobiotin injection into a vocally active VPP site that was part of our pharmacological experiments (see later section) showed retrograde neurobiotin labelling of both PAGcs and PAGcd neurons (Fig. S1). The caudal zones are distinguished, however, by connectivity to the cerebellum (predominantly PAGrs), midbrain tectum (predominantly PAGcs, PAGcd) and midbrain auditory torus (PAGrs, PAGcs) (Fig. S2), and as reported below, patterns of c-fos mRNA expression and pharmacological manipulations.

Compared to silent animals, the number of c-fos mRNA expressing neurons was significantly elevated by two or more-fold in each caudal PAG zone of males making either call type, and in the PAGcs of agonistic versus courtship humming males (Fig. 2i, n–p). The PAGrs exhibited no significant differences in c-fos expression (Fig. S3), leading us to focus subsequent analyses on the caudal zones. The number of courtship and agonistic vocalisations was positively correlated with c-fos+ cell number in the PAGcs (Fig. 2j,k), as were courtship but not agonistic vocalisations with the PAGcd (Fig. 2l,m). We also found greater activation of glutamatergic compared to GABAergic c-fos+ cells in both caudal zones during courtship (~75%) and agonistic (~60%) calling (Fig. S4). Mice similarly exhibit a majority of GLU activation within vocally active PAG neurons[8].

Given the PAG's apparent auditory input (see above), we measured c-fos+ cell number in nesting males that did not hum but were housed with another male that hummed in the same tank for 3-5 days (nests separated by ~15 cm, as in[38]). These group-housed silent males had higher c-fos+ numbers in one PAG zone (PAGcs) relative to isolated silent controls. However, the numbers were significantly lower than in calling males (Fig. 2i), implying that neuronal activation in the PAGcs alone was significantly influenced by audition, but more dependent upon vocalisation. A corollary discharge circuit from the VPP to the auditory sensory epithelium would also decrease a fish's own hearing sensitivity when calling (see[39]).

In aggregate, these findings for a teleost fish demonstrate vocalisation-dependent activation of a PAG region directly connected to forebrain and hindbrain vocal circuitry[15,23,34]. These same characters are basic attributes of a PAG vocal network in mammals[4–10,40].

## Call-specific activation in PAG, but not vocal hindbrain

We next used catFISH, a method that identifies neurons activated by two temporally separated behavioural actions[41,42], to assess if individual PAG neurons were active during specific call types. Following other catFISH paradigms[41,42], there was a 30 min pause between two 5 min behavioural periods (Fig. 3a and "Methods" section). Like earlier studies, we expected that fish calling only during period 1 or 2 (the more recent event) and silent for the remaining 35 mins should predominantly display either c-fos coding cytoplasmic or c-fos intronic nuclear signal, respectively. As predicted, males silent for 35 min and sacrificed immediately after 5 min of agonistic calling (silent-agonistic) had primarily nuclear signal in both caudal PAG zones, whereas the opposite sequence (agonistic-silent) led to mostly cytoplasmic signal (Fig. S5a–f). Agonistic call number during period 2 was positively correlated with c-fos intron cell number in the PAGcs (Fig. S5g, h).

To determine if the same neurons might be involved in only one or both types of calling (courtship, agonistic), we determined the relative degree of neuronal reactivation (i.e., having both nuclear and cytoplasmic signals) between two 5 min periods of the same or different vocalisation types separated by 30 mins (Fig. 3a–d). Although males would not spontaneously hum within 30 min after agonistic calling (see "Methods" section), well beyond the catFISH timeframe (Fig. 3a), they almost immediately made agonistic calls in response to the 3D model during period 2 after courtship humming during period 1 ceased (Fig. 3d, also see "Methods" section). Relative to males that were silent for an entire 40 min trial (i.e., during both periods), the number of cells showing c-fos intronic expression was significantly higher in males that made courtship or agonistic calls during behavioural period 2 (Fig. 3f). Compared to silent controls, fish that produced either courtship or agonistic calls during both periods (Fig. 3b, c) showed relatively high neuronal reactivation (~90%, PAGcs; ~65%, PAGcd; Fig. 3g, h–m). These percentages were significantly greater compared to courtship-agonistic calling males (~65%, PAGcs; 40%, PAGcd; Fig. 3g, n, o). Importantly, the latter further informs us that ~35% of PAGcs and ~60% of PAGcd neurons with intron signal alone were associated only during the most recent behaviour—agonistic calling, suggesting call-specific neuronal activation in the caudal PAG. These findings also suggest that a majority of vocally active PAGcs (~65%) neurons were associated with both vocal states, whereas PAGcd may have more call-specific neurons (~60%). Video analysis further demonstrated that the time spent swimming during agonistic calling was not correlated with c-fos intron expression in the PAG (Fig. S5i, j).

To investigate a possible non-vocal function for vocally active PAG neurons, we examined their participation in foraging (see "Methods" section). Involvement of the PAG in similar behaviours is also implicated in mammals[43,44]. Males in a second catFISH study foraged in two behavioural periods separated by 30 mins or only once during period 1 followed by agonistic calling during period 2 (Fig. 3p, q). Compared to males that foraged twice, c-fos intronic expression was significantly higher by 3.5 (PAGcs) to 10 (PAGcd) fold in males that made agonistic calls during period 2 (Fig. 3r). Thus, agonistic vocalisation was associated with greater neural activity in the caudal PAG compared to foraging. About 75% of these PAGcs neurons, but <5% of the PAGcd neurons, were reactivated when animals foraged twice (Fig. 3s). This suggests that caudal PAG neurons involved in vocalisation and foraging are largely independent populations. Notably, these and the above catFISH results further support a functional distinction between the caudal PAG zones and suggest greater vocal specificity in the PAGcd.

Finally, we used catFISH to investigate if VPP, the lateral PAG's main target within the vocal hindbrain circuitry (Fig. S1)[15,23,45], is active during courtship and agonistic calling. Unlike the PAG, nearly all (> 90%) VPP neurons expressed both c-fos cytoplasmic and nuclear signals during the courtship-agonistic calling sequence (Fig. S6a–c).

In sum, both caudal PAG zones exhibited neurons activated during both vocal contexts, with some uniquely activated only during agonistic calling. Conversely, a call-specific activation pattern was not found for vocal hindbrain interneurons. By investigating both foraging and swimming, we provide further evidence for vocal-specific neurons within the PAG, as in mice[8].

## PAG excitation drives fictive calls that resemble natural calls

A significant advantage of the vocal system in midshipman and other toadfishes is that we can readily record the vocal motor volley, or fictive call that reflects the highly synchronous activity of vocal hindbrain motoneurons that innervate the paired sonic muscles (Fig. S7a, b)[45,46]. Prior studies of midshipman show that electrical microstimulation in the POA-AH, in some cases together with neuropeptide modulation, can evoke PAG vocal neuron activity and fictive calling[23,47]. To first confirm the general necessity

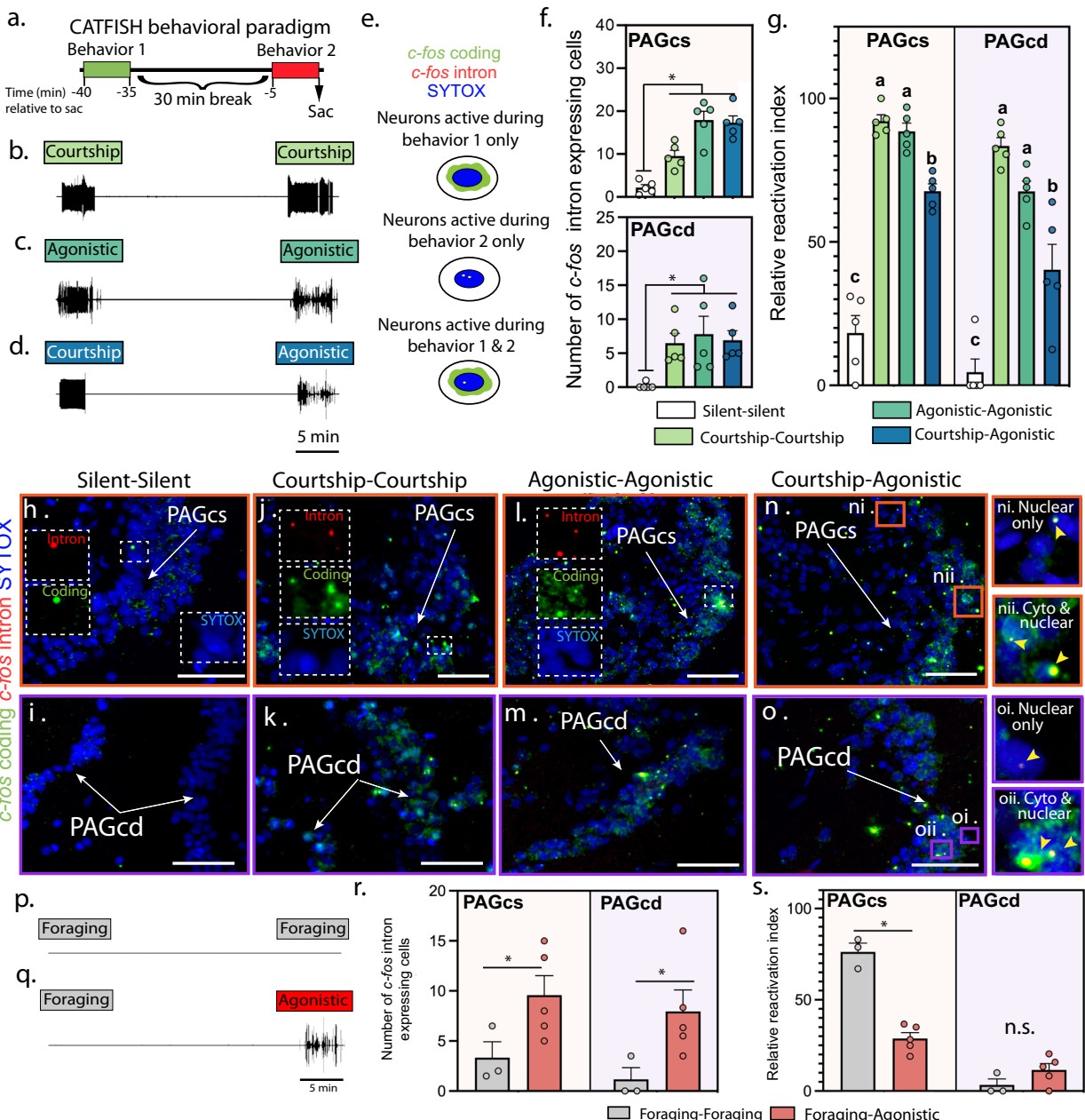

**Fig. 3 | Vocal and non-vocal activation of PAG neurons. a** Experimental timing for catFISH (cellular Compartment Analysis of Temporal activity by Fluorescence In Situ Hybridization) experiments: initial 5 min behavioural period followed by 30 min break and second 5 min behavioural period. **b**–**d** Representative combinations of catFISH behavioural paradigms performed by male midshipman fish ($n = 5$/condition): only courtship (**b**) or agonistic (**c**) calling or courtship followed by agonistic calling (**d**). **e** Schematic of labelling patterns for cells only activated in period 1 (predominantly cytoplasmic) or 2 (predominantly nuclear, intronic), or in both 1 and 2 (nuclear and cytoplasmic). **f** Plots (mean ± SEM) illustrating number of *c-fos* intron expressing cells within the PAGcs (ANOVA: $p = 3.272e-07$) and PAGcd (ANOVA: $p = 1.101e-05$). Asterisk in **f** denotes significant BH corrected post hoc differences ($p < 0.05$, ANOVA). **g** Plot illustrating the reactivation index (percent of cells (mean ± standard error) activated in second behavioural period (nuclear *c-fos* intron signal) that were activated during the first period (cytoplasmic *c-fos* coding signal)) in PAGcs (ANOVA: $p = 6.782e-10$; $n = 5$ animals/condition) and PAGcd

(ANOVA: $p = 1.262e-07$). **h**–**o** Representative confocal images of PAGcs and PAGcd in different behavioural conditions. Dashed white boxes in lower magnification images (**h**, **j**, **l**) illustrate location of inserts showing examples of intron (red), cytoplasmic (green) and SYTOX nuclear (blue) labelling. Orange and purple boxes exhibit examples of nuclear only labelling (**ni**, **oi**; second behaviour only) or cytoplasmic and nuclear labelling (**nii**, **oii**). Scale bars = 250 μm. **n**, **o** Representative combinations of catFISH behavioural paradigms to investigate involvement of vocal PAGcs and PAGcd neurons in a non-vocal behaviour, foraged twice ($n = 4$, **p**) or foraged once followed by agonistic calling ($n = 3$, **q**). **r** Plots (mean ± standard error) illustrating number of *c-fos* intron expressing cells within the PAGcs (two tailed *t*-test: $p = 0.03$) and PAGcd (two tailed *t*-test: $p = 0.01$), $n = 4$/foraging-foraging; $n = 3$/foraging-agonistic calling. **s** Plots (mean ± SEM) illustrating percent of neurons activated in second behaviour period that were also activated during the first period. Asterisk in **r**, **s** denotes significant condition differences ($p < 0.05$, two tailed *t*-test). A detailed summary of statistics is presented in Table S2.

of PAG synaptic activity for vocal network output and that fictive calls are not generated by fibres of passage, we injected muscimol (GABA$_A$R agonist) or a cocktail of two GLU receptor (R) antagonists, NBQX and APV (block AMPA and *N*-methyl-D-aspartate/ NMDA receptors, respectively), into the PAG during extended periods of fictive calling induced by injections of gabazine, a specific GABA$_A$R antagonist[48], into limbic forebrain sites (Fig. S7b–d and "Methods" section). As with primates[49,50], muscimol and

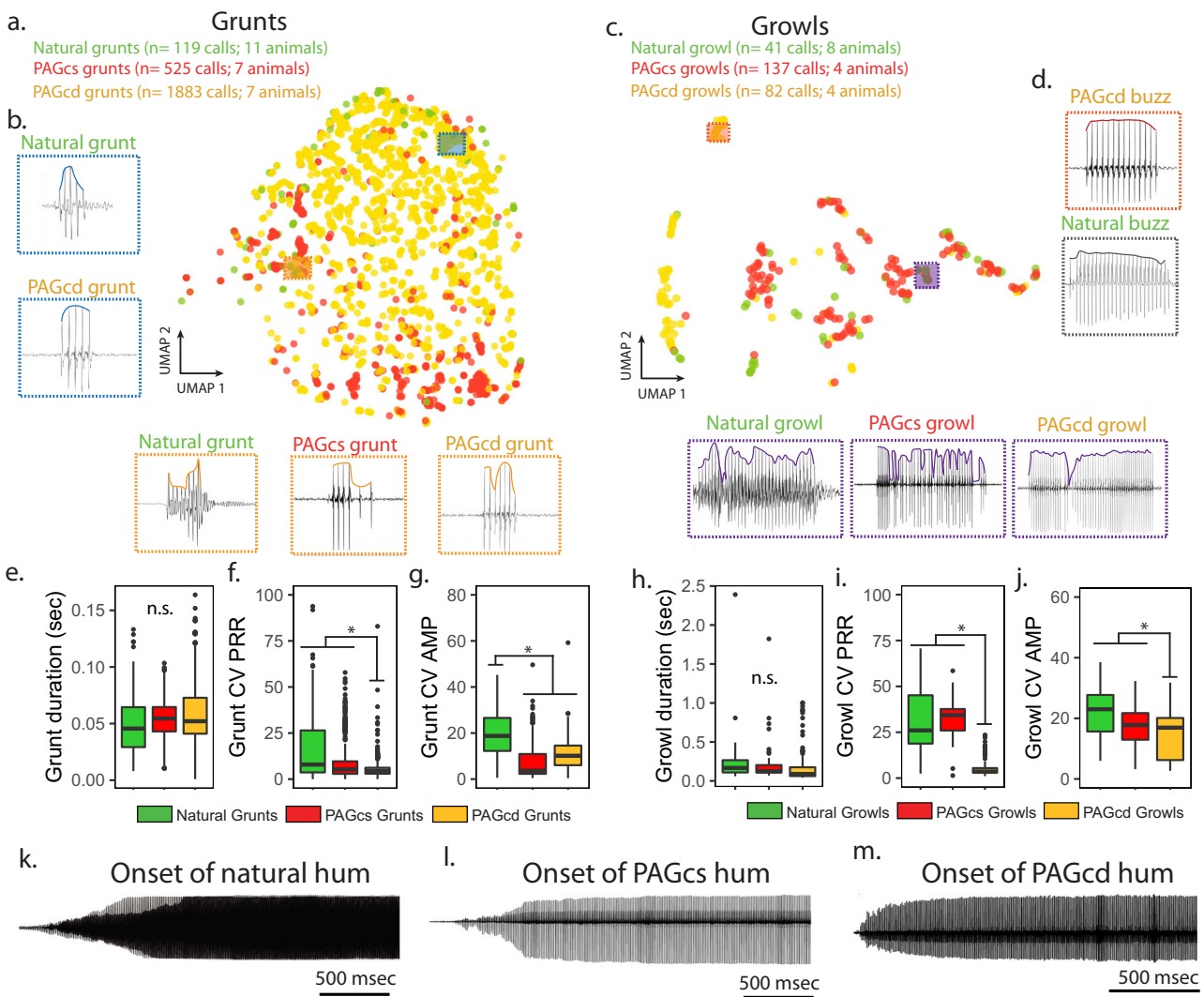

**Fig. 4 | Glutamate (GLU) activation of caudal periaqueductal gray (PAG) generates full range of natural-like fictive calls. a–d** Similar acoustic features between natural and fictive calls were measured, and dimensionality was reduced using UMAP (Uniform Manifold Approximation and Projection) analyses. UMAP plots illustrating how recordings of fictive grunts (**a, b**) or growls (**c, d**) evoked following GLU microinjections in the PAGcs (red) or PAGcd (yellow) overlap natural calls (green) elicited during simulated intrusions using a 3-D model midshipman (Fig. 2g, h). Boxes illustrate representative recordings of natural and fictive calls also shown that were adjacent to each other on UMAP plots. **e–j** Statistical comparisons and box plots of key acoustic features that define grunts and growls, including duration (**e** [grunt: LMM: $p = 0.39$], **h** [growl: LMM: $p = 0.52$]), coefficient of variation (CV) in pulse repetition rate (PRR; **f** [grunt: LMM: $p = 0.02$], **i** [growl: LMM: $p = 0.008$]), and amplitude (AMP; **g** [grunt: LMM: $p = 0.0003$], **j** [growl: LMM:

$p = 0.01$) between natural (grunt: $n = 119$ calls across 11 animals; growl:41 calls across 8 animals), GLU-evoked PAGcs (grunt: 525 calls across 7 animals; growl:137 calls across 4 animals), and GLU-evoked PAGcd (grunt:1882 calls across 7 animals; growl: 82 calls across 7 animals). In box plots, the centre line indicates the median, the edges of the box represent the first and third quartiles, and the whiskers extend to span a 1.5 interquartile range from the edges, and individual dots are points that fall outside this range. *denotes significant post-hoc differences ($p < 0.05$, from linear mixed model analyses); n.s. denotes no significant difference between groups. **k–m** Representative example of a natural courtship hum (**k**) and those evoked following GLU injection into PAGcs (**l**) or PAGcd (**m**). All fictive hums exhibited similar features to natural hums. A detailed summary of statistics is presented in Table S2.

the GLU-R antagonist cocktail greatly or completely inhibited fictive calling (Fig. S7e, f).

To further test for differential involvement of each caudal zone in the patterning of only one or both call types, as suggested by the catFISH results (see above), fictive bouts of calling were evoked by GLU or gabazine microinjections centred in either the PAGcs or PAGcd (Figs. S8 and S9a, b, e, f). GLU-evoked fictive bout duration was similar between zones, although PAGcs bouts had the shortest response latencies, as brief as 0.5 seconds (Fig. S9c, d). To test how well individual fictive calls mirror natural calls, we compared the three most salient temporal features distinguishing call types (Fig. 2a–c)[28,30,31] and which figure prominently in eliciting positive phonotaxis[51] (see "Methods" section): duration, variability in pulse repetition rate (PRR,

equals fundamental frequency of multi-harmonic courtship hums) and variability in amplitude (AMP). We first reduced the dimensionality of these features for each type of GLU-evoked agonistic call using UMAP analyses and plotted the resulting values. Grunts formed a large continuous cluster with all regions of natural and fictive calls exhibiting similar features (Figs. 4a, b and S10). A separate statistical analysis demonstrated that GLU-evoked fictive grunt durations from either caudal zone were similar and overlapped natural grunt durations, although only PAGcs-evoked grunts mirrored natural PRR variability (Fig. 4e–g). Amplitude variability was similar for both zones, but less than that for natural grunts.

Unlike grunts, the more acoustically complex growls formed multiple clusters, each with distinct features (Fig. 4c, d, bottom row

and S11). Regions with large overlap between GLU-evoked fictive growls from both caudal zones and natural growls were often characterised by substantial PRR and AMP variability (Fig. S11). Fictive growls, like grunts evoked from either caudal zone, overlapped natural growl durations, while PAGcs-evoked growls overlapped both natural growl PRR and AMP (Fig. 4h–j). Consistent with this, three clusters that were predominately PAGcd-evoked growls tended to lack the dramatic PRR and AMP variability that characterises natural growls and, hence, exhibited almost no overlap with them (Fig. 4c, far left and S11a). Oscillogram traces, however, revealed a resemblance to natural 'buzzes', which have subtle shifts in AMP and durations like growls and hums (Fig. 4d, top right). Buzzes have only been reported during handling stress, e.g., when collected from a nest[28], although they were not observed during our agonistic trials. Importantly, buzz-like fictive calls further showed the capacity of caudal PAG neurons to generate the full complement of known male sounds.

Although fictive hum-like calls evoked from the PAGcs ($n = 2$) and PAGcd ($n = 4$) were too few for similar quantitative analyses, each one closely resembled natural hums, including the gradual increase in AMP at the beginning of fictive hums followed by stable PRR and AMP[28–31] (Figs. 2a–c and 4k–m). The greater number of fictive hum and buzz-like calls elicited from the PAGcd suggested a more salient role in suppressing variability, i.e., stabilising both PRR and AMP. A more significant role for PAGcd in courtship humming is also consistent with the positive relationship between the number of hums, but not agonistic calls, produced and c-fos expression in the PAGcd (Fig. 2l).

Finally, we checked whether GLU volume, and by extrapolation the amount of activation, played a role in evoking a particular call type. Increasing GLU volume at the same PAGcs site altered neither the latency nor the types of fictive calls evoked (Fig. S12). This provided further support for concluding that activation of specific PAG regions, and not the extent of activation, drives context-specific vocal output.

## Blocking GABAergic action in PAG generates persistent agonistic-like vocal output

We next wanted to test the effect of reducing local inhibition on fictive call output from the PAG. Work in mice suggests that disinhibition in the PAG promotes louder and longer USV bouts, but it is unknown whether this influences the temporal patterning of different call types[6–8]. Gabazine injection within the midshipman PAGcs and PAGcd only evoked agonistic-like fictive calls (Figs. 5 and S9e, f). This included minute-long bouts (Fig. S9g), a feature of natural bouts of grunts and growls[30], and monkey vocalisations evoked with a $GABA_AR$ antagonist[49]. Also like mouse and monkey[6,49], removing local inhibition in the caudal zones led to much longer response latencies compared to GLU action (Fig. S9d, h).

The UMAP plots showed that gabazine-evoked fictive grunts formed a single large cluster that overlapped natural grunts (Figs. 5a and S13a), whereas gabazine-evoked fictive growls formed several clusters having substantial overlap with natural calls (Figs. 5b and S14a). The largest growl cluster was defined by substantial PRR variability, a defining feature of natural growls (Fig. 5b, top right and S14a, f–h)[29,30]. The three bottom clusters, however, exhibited no overlap with natural growls, tending to be longer in duration with minimal variability in both AMP and PRR (Fig. S14c–h). Instead, these responses resembled buzzes (Fig. 5b, bottom), like some GLU-evoked PAGcd responses (Fig. 4d).

Statistical analyses demonstrated that individual gabazine-evoked fictive grunts from the PAGcs zone overlapped the duration and from both caudal zones the PRR variability of natural grunts; neither zone's responses overlapped natural AMP variability (Fig. 5c–e). Gabazine-evoked growls from both caudal zones only overlapped natural growl PRR variability (Fig. 5f–h).

In sum, GLU or gabazine actions in the caudolateral PAG of midshipman fish suggested that separate zones may differentially bias production of different acoustic features and, in turn, call type. Thus, for the 12 total comparisons made between the acoustic features of natural and GLU- or gabazine-evoked fictive grunts (6 comparisons, Fig. 4e–j), and natural and GLU- or gabazine-evoked fictive growls (6 comparisons, Fig. 5c–h), the PAGcs and PAGcd fictive calls overlapped natural call features in 9/12 and 4/12 instances, respectively (Table S1).

## Vocal hindbrain gating of context-specific vocalisation

We next wanted to know if the ability of the PAG node to evoke fictive calls that closely resembled natural context-specific vocalisations is shared with its vocal hindbrain target, the VPP (Fig. S1a, b). We focused on fictive calls evoked by GLU microinjection (Fig. 6a, b) given the evidence for monosynaptic excitatory input from the midbrain onto VPP neurons[45], extensive GLU input onto VPP somata (Fig. 6c), and GLU's potent role in evoking mammalian natural calls[49] and midshipman fictive calls (see above). Nearly half of the GLU-evoked VPP responses were agonistic-like, with several fold more grunt (42%) than growl-like (12%) responses; the remainder were one pulse, unlike any natural or PAG-evoked calls. Hum-like calls were never evoked. Like the PAGcs (Fig. S12), varying amounts of GLU had little, if any, apparent effect on responses (Fig. S15). Compared to the PAG, VPP responses had much shorter durations that resembled natural grunts and shorter response latencies following GLU microinjection (Fig. 6d, e), consistent with VPP's extensive connectivity to the pacemaker (VPN)–motoneuron (VMN) circuit that directly activates vocal muscles[45,46]. While natural grunts can occur singly, they more frequently occur at a relatively invariant inter-grunt interval (IGI) in bouts known as grunt trains during aggressive encounters[28–30]. The temporal structure of fictive bouts of sound pulses evoked from VPP tended to be erratic compared to natural and PAG-evoked responses (Fig. 6f). In particular, unlike the stable IGI of natural grunt trains and fictive ones evoked with comparable GLU injection volumes into the PAGcs, the IGI of VPP-evoked bouts of successive grunts tended to increase and then decrease over the course of the responses (Fig. 6g, h).

As with the PAG, we statistically compared acoustic features between single natural grunts and GLU-evoked fictive grunt and growl-like responses (Fig. 6i–p). All VPP features were reduced and the resulting UMAP values plotted, but we only analysed naturalistic ones having more than one pulse (see above). There was significant overlap for all features between VPP-evoked fictive and natural grunts (Fig. 6k–m). In contrast, fictive VPP growls exhibited significantly longer durations and more variable PRR compared to natural growls (Fig. 6n–p).

In aggregate, the analyses for VPP strongly suggested that the temporal patterning of natural grunt trains, along with individual growls and hums, requires PAG activation.

## Discussion

This study, the most comprehensive to date of vocally active midbrain neurons in a non-mammal, provides behavioural, cellular and neurophysiological evidence that a teleost fish shares multiple characters of a vocal PAG node with mammals. This includes evidence for the sufficiency of GLU-dependent excitation within the PAG to evoke the full range of social context-specific call types and the activation of individual caudolateral PAG zones during the production of unique vocalisations (see below). Also like mammals[25,49,52–54], evidence from the current and prior studies[15,23,34] of direct PAG connectivity with vocal hindbrain and limbic forebrain sites, our findings provide support for functionally similar vocal networks shared between two vertebrate lineages separated by millions of years of evolution.

The vocal PAG node in midshipman fish also shares several neurophysiological characters with mammals. For example, single neuron activity in the PAG of primates is correlated with the production of one or more call types[55]. Similarly, our c-fos behavioural experiments show that while some vocal PAG neurons appear to be associated with only

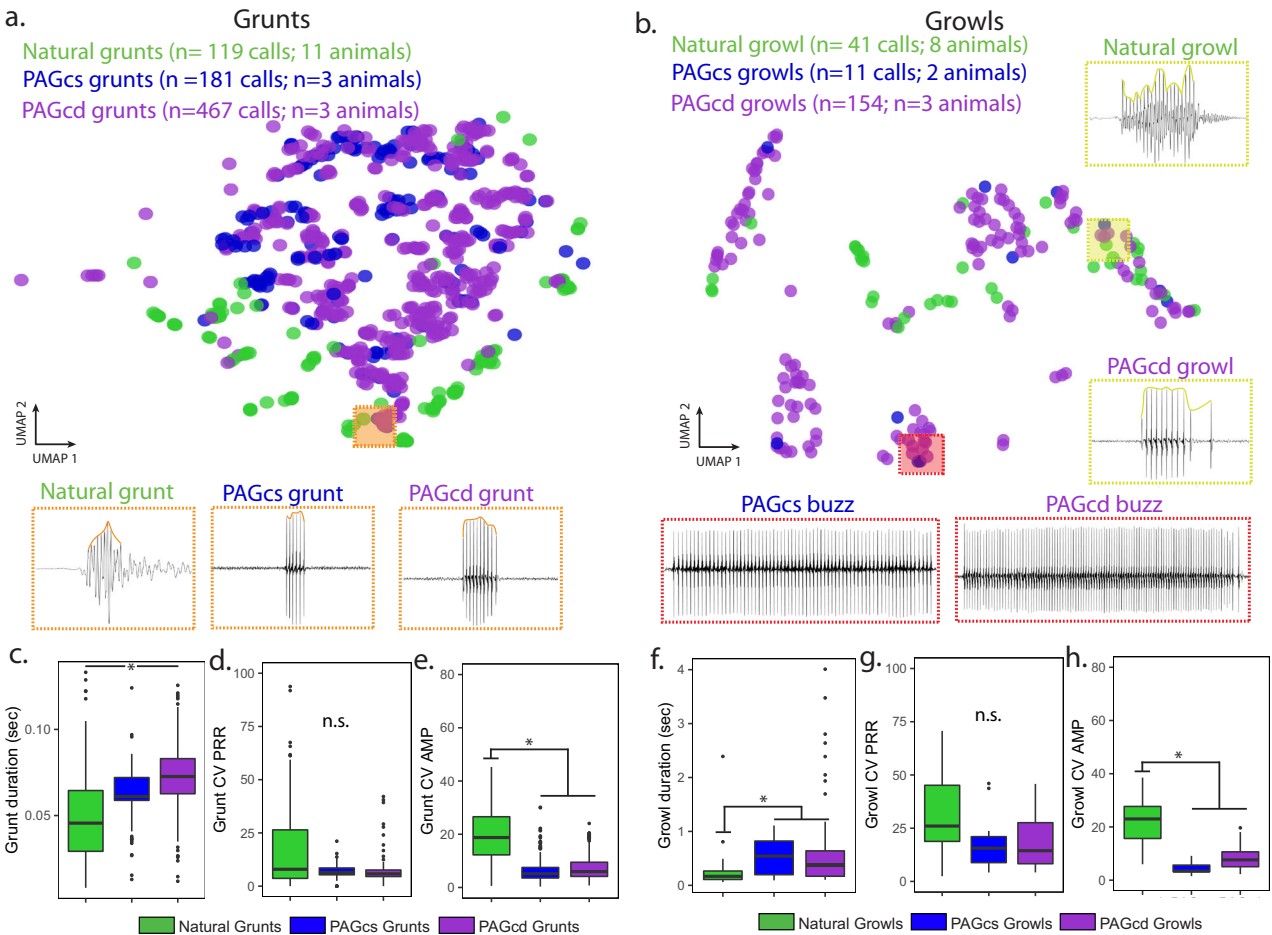

**Fig. 5 | Removing inhibition of caudal periaqueductal gray (PAG) neurons does not generate full range of natural-like fictive calls. a, b** Similar acoustic features between natural and fictive calls were measured, and dimensionality was reduced using UMAP (Uniform Manifold Approximation and Projection) analyses. UMAP plots (top) illustrating how gabazine-evoked fictive grunts (**a**) or growls (**b**) from the PAGcs (blue) or PAGcd (purple) overlap natural calls (green). Orange boxes illustrate representative recordings of natural and fictive calls also shown that were adjacent to each other on the UMAP plots. **c–h** Statistical comparisons and box plots of key acoustic features that define grunts and growls, including duration (**c** [grunt: LMM: $p = 0.02$], **f** [growl: LMM: $p = 0.04$]), coefficient of variation (CV) in pulse repetition rate (PRR; **d** [grunt: LMM: $p = 0.40$], **g** [growl: LMM: $p = 0.61$]), and

amplitude (AMP; **e** [grunt: LMM: $p = 0.0008$], **h** [growl: LMM: $p = 0.0006$]) between natural (grunt: $n = 119$ calls across 11 animals; growl: 41 calls across 8 animals), Gabazine-evoked PAGcs (grunt: 181 calls across 3 animals; growl: 11 calls across 2 animals), and Gabazine-evoked PAGcd (grunt: 467 calls across 3 animals; growl: 154 calls across 3 animals). In box plots, the centre line indicates the median, the edges of the box represent the first and third quartiles, and the whiskers extend to span a 1.5 interquartile range from the edges, and individual dots are points that fall outside this range. *denotes significant post-hoc differences after BH correction ($p < 0.05$); n.s. denotes no significant difference between groups. A detailed summary of statistics is presented in Table S2.

agonistic calling, others are active during both agonistic and courtship calling. Consistent with indications of GLU- and GABA-dependent activation of different call types in monkey and cat PAG[25,49,54], pharmacological manipulations in midshipman PAG suggest that reducing local GABA-dependent inhibition could contribute to activating agonistic (grunts or growls), but not courtship calling. Also, like recordings in monkeys suggesting a PAG role for influencing call features[52], reduced inhibition in either caudal PAG zone leads to significant increases in the duration and AMP stability (i.e., less variability) of individual agonistic-like, fictive calls (Table S1). The greater number of fictive hums evoked from the PAGcd together with the observed pharmacological effects in the PAGcd suggest that selectively increasing excitation (GLU effects) and decreasing inhibition (gabazine effects) in this zone could lead to decreased PRR and AMP variability (e.g., Figs. 4f, g, i, j and 5e, h) along with increased call duration (e.g., Fig. 5c, f) respectively, thereby enhancing the key acoustic features of courtship calls (Fig. 4k–m)[28–30].

Findings for mammals similarly imply divergent roles for vocal regions of the PAG and hindbrain. While a study of cats proposed the

PAG as a vocal pattern generator site[54], we provide the strongest evidence for any vertebrate that PAG and not hindbrain neurons can influence both the fine structure and temporal envelope (PRR, duration, amplitude) of acoustically complex vocalisations. The PAG's target within the midshipman's vocal hindbrain circuit, the VPP, apparently has the capacity to generate simple grunts, but not the more complex agonistic (growls, grunt trains) and courtship calls evoked from the PAG (Fig. 7a). Vocally active PAG, but not VPP, sites could produce the temporally stable silent intervals between successive grunts that characterise grunt bouts (trains), comparable to inter-syllable intervals in avian vocalisations and rodent USV bouts[8,32]. Moreover, only VPP-evoked responses overlapped with the call duration, PRR and AMP of natural grunts, consistent with evidence showing a prominent role for the vocal hindbrain circuit in determining the fine structure and temporal envelope of individual grunts (Fig. 7a)[45]. Similarly, monkeys have hindbrain populations whose activity correlates with a syllable's fine and gross temporal structure[56]. In cats, pharmacological manipulation in nucleus retroambiguus, a PAG hindbrain target that innervates expiratory motoneurons, largely

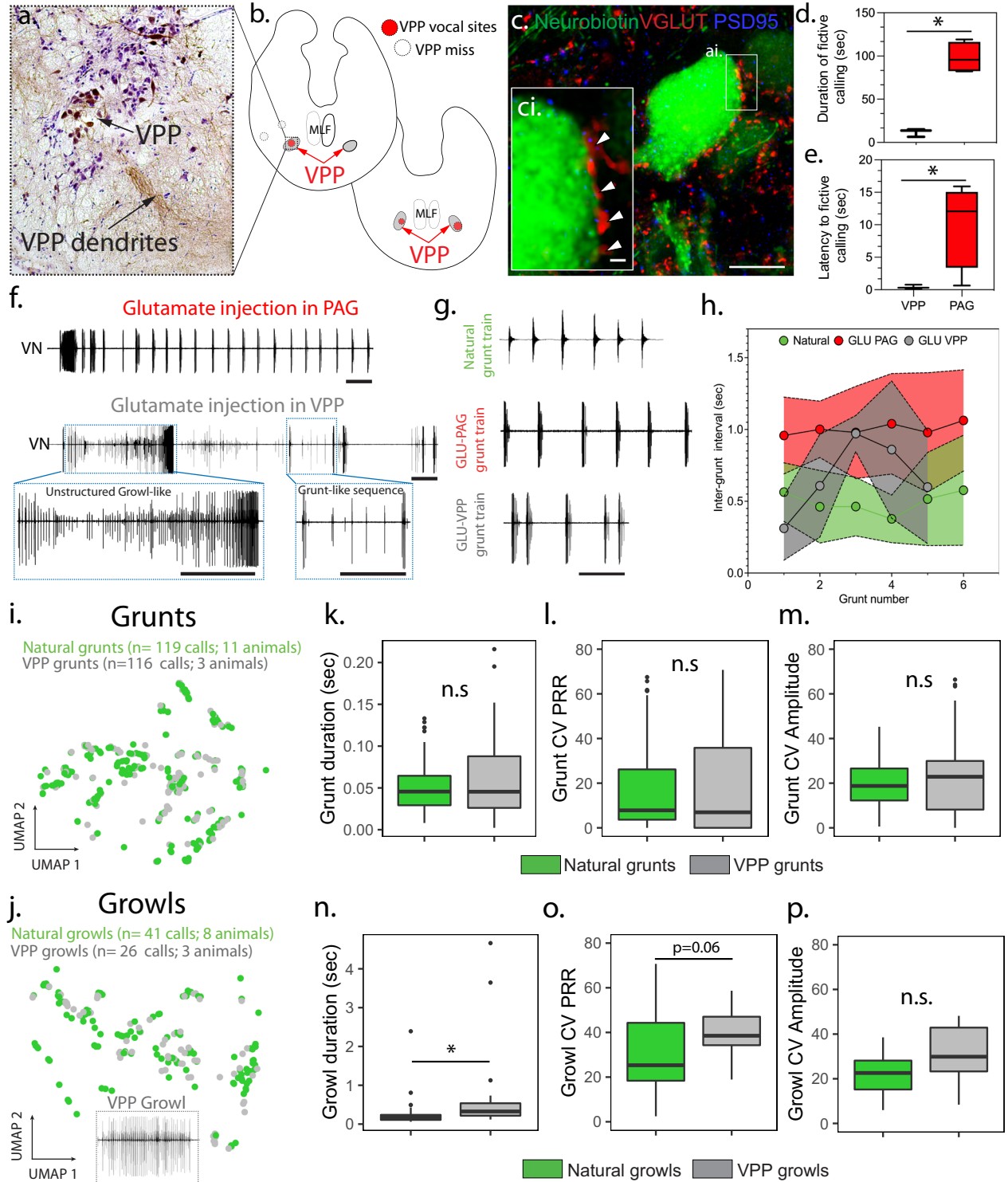

evokes unmodulated tonal sounds[25,57], whereas PAG stimulation evokes acoustically complex calls (e.g., mews, howls)[25,54]. Electrical stimulation in rat hindbrain only evokes single USVs (single grunt analogue) that do not bear a strong resemblance to natural calls (see Fig. 1b in ref. 58), whereas PAG stimulation evokes USV bouts[58] (grunt bout analogue, also see [6–8] for mouse PAG).

Collectively, the evidence points to a PAG vocal network in fishes and mammals that shares multiple characters (Fig. 7). This includes a fundamental role of the hindbrain in refining the properties of basic acoustic units (e.g., note, syllable, grunt) and of the PAG in selectively

reconfiguring the output of one or more of these units, leading to a range of temporally-rich, social context-specific call types (e.g., songs, advertisement calls, long bouts/trains) (Fig. 7). This is not to say that vocal networks lack clade- and/or species-specific attributes. For example, compared to fishes, the task of initiating and patterning call types is far more demanding for most tetrapods given the need to coordinate multiple neuromuscular compartments during phonation, especially those underlying respiration (Figs. 1 and 7)[5,8,26,59]. The comparatively strong influence of the PAG on the temporal coding of call types at least in toadfishes like midshipman, may have co-evolved with

**Fig. 6 | Glutamate (GLU) modulation of vocal hindbrain. a** Neurobiotin-filled VPP neurons -75 μm caudal to VPP injection site. **b** Locations of GLU-neurobiotin injections into VPP (red circles) White circles illustrate non-vocal sites that missed VPP (n = 2 animals) and showed no label in VMN or PAG. **c** Super-resolution image of neurobiotin-filled VPP neurons labelled with VGLUT (red) and a post-synaptic marker for glutamatergic neurons (PSD-95, blue). (**ci** is high magnification of region outlined withwhite box in **c**). Scale bars = 10 μm (**a**), 1 μm (**ai**). **d, e** Box plots comparing fictive call duration (**d**; two-tailed *t*-test: *p* = 0.0003) and latency to start calling (**e**; two-tailed *t*-test: *p* = 0.04) following GLU-VPP injection (n = 3) or periaqueductal gray (PAG) (n = 4). * denote significant differences. **f** Fictive calls recorded from vocal nerve (VN) following GLU injection in PAG (top) or VPP (bottom). Blue boxes in upper record indicate location of lower records of unstructured growls and grunt trains produced following GLU-VPP injection compared to structured ones following GLU-PAG injection. Scale bars = 1 s. **g** Natural, GLU-PAG

and GLU-VPP bouts of grunts. Scale bar = 1 sec. **h** Average inter-grunt interval (±SD) for natural and GLU evoked bouts of grunts (5–7 grunts/bout; n = 6 natural; 7 GLU-PAGcs; 4 GLU-VPP). **i, j** UMAP plots illustrating how individual GLU-VPP (grey), fictive grunt-like (**i**) and growl-like calls (**j**) overlap natural calls (green). **k–p** Statistical comparisons and box plots of key features defining grunts and growls: duration (**k**[grunt: LMM: *p* = 0.47], **n**[growl: LMM: *p* = 0.01]), coefficient of variation (CV) in pulse repetition rate (PRR; **l**[grunt:LMM: *p* = 0.59], **o**[growl: LMM: *p* = 0.06]), and amplitude (AMP; **m**[grunt: LMM: *p* = 0.38], **p**[growl: LMM: *p* = 0.17]) between natural (grunt n = 119 calls /11 animals; growl n = 41 calls/ 8 animals) GLU-VPP (grunt: 116 calls/3 animals; growl:26 calls/3 animals). Box plots: centre line indicates median edges represent first and third quartiles; whiskers extend to span a 1.5 interquartile range from edges; individual dots are points falling outside range. *denotes significant post-hoc differences (*p* < 0.05); n.s. denotes no significant difference between groups.

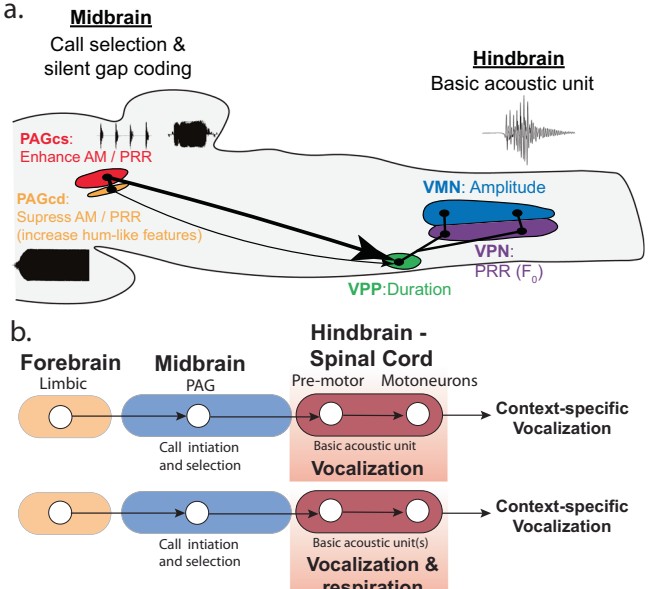

**Fig. 7 | A conserved midbrain periaqueductal gray (PAG) node for social context-specific vocalisation. a** Sagittal brain drawing for midshipman fish summarises proposed roles of the caudal superficial (cs) and deep (cd) PAG zones in generating social context-specific vocal output (current study), and how hindbrain-spinal vocal prepacemaker (VPP), pacemaker (VPN), and motor (VMN) nuclei shape acoustic features of grunts (see [45]). **b** PAG stimulation via glutamate or excitatory amino acids leads to vocal output in fish (current study) and mammals (see Discussion). Schematic drawing of proposal that the PAG activates one or more hindbrain-spinal motor populations driving vocalisation, including those for respiration, a major character distinguishing mammals from fishes given the dependence of vocalisation among most tetrapods on respiration. Hindbrain-spinal motor populations in toadfishes (e.g., midshipman fish) and mammals (e.g., see [56,73]) are proposed to refine the properties of clade- or species-typical basic acoustic unit(s), with the PAG selectively reconfiguring the output of one or more units to determine the temporal features of different context-specific vocalisations.

a relatively simple sonic mechanism, namely patterning the output of only a single hindbrain circuit that innervates one pair of sonic muscles. The evolution of call types and central networks among tetrapods[1,3,5,40,60] (Fig. 1) may depend, in turn, upon a PAG node that provides timing signals to coordinate the final output of multiple clade- and/or species-typical hindbrain units.

## Methods

### Animals

Adult type I male midshipman fish were collected from nests in northern California during the breeding season and shipped to Cornell University where they were housed in environmental control rooms in

artificial seawater tanks at 16−18 °C on a 15 h: 9 h, light: dark cycle as to mimic natural long days of the breeding season[31]. Males were provided artificial nests and housed individually. For all conditions, except the silent and hearing courtship calls (see below), animals were kept in 25 gal tanks. All animals were acclimatised for at least 3 days before performing behavioural experiments. Animals selected for behavioural experiments varied slightly in body size (15.2–18 cm), but there were no significant differences between courtship and agonistic calling animals (p > 0.2) or vocal animals and controls (p > 0.2). As previously described[20,31], vocalisations were monitored remotely using hydrophones (Aquarian Audio H1a, Aquarian Audio and Scientific, Anacortes, WA; frequency response of 20 Hz to > 100 kHz). Methods for fixation and tissue sectioning are described in detail elsewhere[20]. All research complies with all relevant ethical regulations and all procedures were approved by the Institutional Animal Care and Use Committee of Cornell University. The capture of midshipman fish was approved by CA Fish and game with the assistance of personnel at the UC Davis Marine lab.

### Behaviour experiments

**Courtship vocalisation.** For *c-fos* mRNA mapping across the vocal motor network (n = 8) and identifying the neurotransmitter phenotype of *c-fos* expressing PAG neurons (n = 5), observers remotely monitored humming (one or multiple hum bouts) over the course of a 40 min trial after the onset of the first hum. CatFISH experiments were performed after males acclimated to aquaria for at least 2 weeks. Males that spontaneously hummed[20,31] on previous days were monitored to ensure that we could capture them within the catFISH time parameters used in other studies[41,42]. It was not possible to have exact 30 min pauses between two hum periods because disruption (removal of nest cover/roof or simulated intrusion, see below) to a humming male rarely resulted in additional hums. However, when males first hummed each subjective night, they often produced short bouts followed by pauses. We took advantage of this by capturing animals (n = 5) that paused 25–35 min between their first (behaviour 1 for catFISH) and second (behaviour 2 for catFISH) hums, similar to the timeframe described in catFISH experiments[41,42]. Variation in this pause did not influence the amount of *c-fos* mRNA expressing cells in PAGcs or PAGcd (p > 0.2).

**Agonistic (grunt, growl) vocalisation.** To perform simulated resident - intruder experiments, we used a silicon 3D printed male midshipman (20 cm, standard length) model that was generated through laser scanning an adult nesting male midshipman. Pilot experiments demonstrated that when male midshipman encountered this model, they responded with upwards of 200 vocalisations in 10 min, and that this behaviour was highly repeatable. For these behavioural trials (*c-fos* mRNA mapping: n = 6; GLU/GABA phenotype identification: n = 5; catFISH: n = 5), as with humming males, we ensured that males were inside their nest. To provoke agonistic vocalisations, we attached the

model to a transparent fishing line that allowed us to manually move it for 30 min. At the start of the trial, the model was always brought up to the nest entrance. Eventually, the male would leave the nest, and we would then chase it around the tank. During these trials, males would occasionally return to the nest. All vocalisations were recorded using a hydrophone. For catFISH experiments, we (1) evoked agonistic calling for 5 min with the model during behavioural period 1 followed by a 30 min break of no calling and then a second 5 min period of model-evoked agonistic calling for 5 min, or (2) recorded spontaneous courtship humming during behavioural period 1 followed by a 30 min break of no calling and then a second 5 min period of model-evoked agonistic calling. Each 5-min agonistic calling period was video recorded (Canon Vixia HFR500), call number assessed by visualisation of waveforms in Audacity (v2.0.6) by an observer blind to the condition (e.g., agonistic calling only, agonistic-silent, silent-agonistic, agonistic-agonistic, or courtship-agonistic). All fish in the agonistic catFISH experiment produced agonistic calls throughout each 5-min period. The total number of agonistic calls produced across animals varied (range: 2 – 21 grunts; range: 0 – 16 growls) and thus allowed us to investigate how *c-fos* expression was associated with the amount of agonistic calling.

**Courtship-agonistic calling.** For these catFISH experiments ($n = 5$), we remotely monitored nests of males that had produced bouts of courtship hums on the previous day. Males were allowed to hum for up to 5 min during behavioural period 1, after which the terracotta lid to their nest was removed (see Fig. 1d). Initial trials determined that this was necessary to prevent the fish from continuing to produce courtship hums. After a 30 min pause, we returned and provoked agonistic vocalisations with the 3D model fish for 5 min during behavioural period 2. Videos of the second behavioural period, agonistic calling, were recorded and call number assessed by an observer blind to the behavioural condition as described above.

When the terracotta lid was removed, males would remain buoyant for a short period of time due to swim bladder inflation during humming. By the time the agonistic trials began, the fish no longer had the swim bladder fully inflated. We performed a pilot study showing that lid removal did not alter the total amount of agonistic vocalisations that males produced ($n = 6$ with no lid removed; $n = 4$ with lid removed).

**Silent controls.** Silent controls were also removed from their home aquarium during the dark period. As above, vocal activity was monitored with hydrophones. This allowed us to confirm that there was no vocal activity in the 2 h timeframe prior to sacrificing these control animals (*c-fos* mRNA mapping: $n = 5$; catFISH: $n = 5$).

**Group-housed silent males exposed to hums.** A second set of controls investigated the degree to which hearing a hum might contribute to activation of PAG neurons ($n = 3$). Two males were individually housed with an artificial nest identical to other experimental conditions in larger tanks split with a mesh divider (see Fig. 3A in [38]). We remotely monitored each male with separate hydrophones placed near their respective nests, allowing us to ensure that the male hearing hums did not produce any vocalisations.

**Foraging.** Males will often leave their nest during the dark period to capture prey (nesting males gain weight and feed[61,62]). They were fed small "feeder" goldfish about once a week before behavioural experiments (part of their protocol approved diet). During these feeding trials, most males would remain in the nest (see Movie S3) and occasional swim to capture the goldfish. We ensured that all males ate the same number of fish in each trial. For example, fish in the foraging-foraging condition ate one fish during each 5 min behavioural period (1 and 2). Conversely, fish in the foraging-agonistic sequence ate only one

fish in behavioural period 1. Thus, for these catFISH analyses males were fed twice with small feeder goldfish during both behavioural periods or fed once during period 1 and then induced with the 3D model to produce agonistic calls during period 2 (see agonistic vocalisation experiment above). As in all catFISH experiments, the first feeding period was separated from the second behaviour, foraging ($n = 3$) or agonistic calling ($n = 5$), by a 30 min break (see Fig. 3a). For all experiments, we both visually confirmed that animals foraged and examined stomach contents to confirm that they fully ate each goldfish provided to them.

### Analysis of swimming during agonistic calling catFISH trials

To investigate whether the amount of swimming ($n = 8$) influenced PAGc *c-fos* intron expression we analysed videos in Behavioral Observation Research Interactive Software (BORIS[63]). Since *c-fos* mRNA expression can remain expressed in the cytoplasm for ~50 mins after a behaviour has occurred[41,64], we restricted this analysis to *c-fos* intronic expression. In this way, any expression would only be influenced by behaviour produced in the last 5 min (e.g., second period of agonistic calling). An observer blind to the conditions, scored swimming when there was visually detectable swimming in any direction. In BORIS this was coded as an event with onset and offset times. These events were summed up to get a total time swimming during the 5-min period.

### Cloning, partial sequencing and riboprobe synthesis of *c-fos*

To identify the *c-fos* sequence in midshipman, PCR primers (forward primer:GGGACAACCTGGGATACTACC; Reverse primer: TGAGGGGATTTTGCAGATGGG) were generated from ~ 641 bp region of this gene that were highly conserved across teleost fish. The *c-fos* intron was identified by first using c-fos coding region primers sequencing on genomic DNA. Then primers specific first intron of the midshipman *c-fos* gene were designed (forward: GTTGGCTGCACCTCTGAAGA; reverse: TGACCATGGTAGCCTACTGC). PCR was carried out using identical parameters to previous studies from our lab[20].

The purified PCR product was inserted into pCRII-blunt TOPO vector and transformed into TOP10 competent cells via heat shock for 30 s at 42 °C. Cells were then cultured in SOC media for 1 h at 37 °C before being spread on LB agar plates with 50 mg/ml kanamycin. The next day, individual colonies were transferred to LB liquid culture containing kanamycin, allowed to grow overnight, and then purified by using a miniprep kit (Qiagen Miniprep Kit). The resulting minipreps were sent to Cornell's Genomic Facility for sequencing to verify the identity and orientation of PCR product in each plasmid.

Plasmids were linearised with EcoRV or SpeI restriction enzymes. In vitro transcription of antisense probes was carried out in a reaction containing 0.1 M DTT, 2.0 µl Transcription Buffer, RNase Inhibitor, SP6 Polymerase, 10x DIG or FITC Labeling Mix (40 U/µl), and the 1 µg of linearised midshipman *c-fos* coding (FITC or DIG; *see below*) or *c-fos* intron (DIG) plasmid. Probes were then precipitated overnight in isopropyl alcohol and lithium chloride at −20 °C, centrifuge at $14,000 \times g$ for 30 min, and reconstituted in 90% formamide in nuclease-free water.

### Fluorescent in situ hybridization

At the end of the behavioural experiments, fish were deeply anaesthetised in 0.025% benzocaine dissolved in aquarium water, exsanguinated, and then perfused transcardially with ice-cold marine teleost Ringers solution followed by 4% paraformaldehyde (PFA) in 0.1 M phosphate buffer (PB). Brains were immediately dissected, postfixed in 4% PFA for 1 h at room temperature, and then washed in 0.1 M PB at 4 °C overnight. The next day, the tissue was cryopreserved in 30% sucrose in 0.1 M PB and embedded Tissue-Tek O.C.T. compound (Sakura Finetek, Torrence, CA, USA) before being stored at −80 °C. Brains were sectioned at −20 °C in a cryostat into 25 µm transverse sections and thaw-mounted onto Superfrost Plus slides (ThermoFisher

Scientific, Waltham, MA, USA). Adjacent sections were collected as three complete series through the brain. Tissue sections were allowed to dry at room temperature and later stored at −80 °C. Sections were later removed from the −80 °C freezer and first fixed in 4% paraformaldehyde for 5 min and then washed in PBS twice for three min each. Sections were then acetylated for 10 min in a 0.1 M TEA solution with 0.33% acetic anhydride, rinsed once with PBS, and then serially dehydrated in 70%, 95% and 100% ethanol. Sections were next incubated in hybridization buffer (300 mM NaCl, 20 mM Tris-HCl pH 8, 5 mM EDTA, 10 mM Na2HPO4 (pH=7.2), 10% dextran sulfate, 1X Denhardt's, 500ug/mL tRNA, 50% formamide in DI water) at room temperature for 1 h. Slides were then transferred to hybridization buffer with antisense FITC (c-fos mRNA)-labelled riboprobe. Slides were cover-slipped, sealed, and incubated overnight at 60 °C. The following day, sections were rinsed, coverslips removed, and then washed in 5x SSC for 10 min followed by 0.2X SSC three times for 20 min. After rinsing slides in TNT (Tris-NaCl-Tween) buffer, endogenous biotin signal was quenched using by first streptavidin blocking reagent for 30 min (component A; Endogenous Biotin-Blocking Kit; Invitrogen). Next, we performed two 5 min PBS washes before adding the biotin blocking reagent for 30 min (component B; Endogenous Biotin-Blocking Kit; Invitrogen). Slides were then incubated in blocking buffer (0.5% blocking reagent [Roche] and 5% horse serum diluted in TNT buffer) for 1 h at room temperature. Next, slides were incubated in anti-FITC-HRP antibody diluted in TNT buffer (1:1000) overnight at 4 °C. The following day, slides were washed three times in TNT buffer for 10 min each. We then incubated slides in biotinylated-tyramide (1:150; Akoya biosciences) diluted in amplification buffer (Akoya biosciences) for 15 min. We then incubated slides in strepavidin-alexa 488 (1:1000; ThermoFisher) overnight at 4 °C. Finally, slides were washed 3 times in PBS and then mounted in Prolong Gold with SYTOX-deep red. Slides were stored at 4 °C.

### Double fluorescent in situ hybridization (catFISH)

For double FISH, slides were hybridised with FITC (c-fos coding) and DIG (c-fos intron) probes. We followed the protocol for single-coloured FISH until the incubation in biotinylated-tyramide (see above). After this step, slides were washed twice in PBS before incubating slides in 2% sodium azide to quench residual HRP activity. Slides were then washed 3 times in PBS before being immersed in blocking buffer for 10 min at room temperature. We then incubated slides in strepavidin-alexa 488 (1:1000; development of c-fos coding signal) and DIG-HRP (1:1000) overnight at 4 °C. The next day, slides were washed three times in TNT buffer for 10 min each. We then incubated slides in cy3-tyramide (Akoya Biosciences; 1:150) diluted in amplification buffer (Akoya Biosciences; development of c-fos intron) for 10 min. Finally, slides were washed three times in PBS and then mounted in Prolong Gold with SYTOX-deep red. Slides were stored at 4 °C.

### Fluorescent in situ hybridization and immunohistochemistry

Tissue sections were then allowed to dry at room temperature and later stored at −80 °C. Brain sections were later removed from the −80 °C freezer and first fixed in 4% paraformaldehyde for 5 min and then washed in PBS twice for three min each. Sections were then acetylated for 10 min in a 0.1 M TEA solution with 0.33% acetic anhydride, rinsed once with PBS, and then serially dehydrated in 70%, 95% and 100% ethanol. Sections were next incubated in hybridization buffer at room temperature for 1 h. Slides were then transferred to hybridization buffer with antisense DIG (c-fos mRNA)-labelled riboprobe. Slides were cover slipped, sealed, and incubated overnight at 60 °C. The following day, sections were rinsed, coverslips removed, and then washed in 5x SSC for 10 min followed by 0.2X SSC three times for 20 min. Sections were rinsed briefly in 100 mM Tris, pH 7.5; 150 mM NaCl, and then blocked for 60 min at room temperature in blocking buffer (0.1 M Tris, 150 mM NaCl, 10% normal horse serum). Slides were

next transferred to blocking buffer with an HRP conjugated anti-DIG antibody (Roche) an allowed to incubate at 4 °C overnight, and then were washed three times for 10 min in 100 mM Tris, 150 mM NaCl, pH 7.5. The slides were next developed by incubating with Cy3-TSA kit (Akoya Biosciences; 1/150) as described above, washed three times with PBS, and then incubated in blocking solution (0.2% bovine serum albumen, BSA, Sigma-Aldrich, St. Louis, MO), 0.3% Triton-X 100 (Sigma-Aldrich), and 10% normal goat serum (NGS, ThermoFisher) in PBS. Then, sections were incubated overnight with anti-GABA (Synaptic systems;1:500) and anti-GLU (Sigma-Aldrich; G6642;1:500) antibodies in blocking solution at room temperature in a humidified chamber. Previous experiments have extensively validated both antibodies. For instance, the GLU antibody specifically labels neurons expressing a fluorescent protein under control of the vglut2 promoter[65], and preadsorption with glutamate eliminates labelling in another teleost[66]. Likewise, with the GABA antibody, elimination of label following preadsorption of the antibody with GABA has been reported in a broad range of species[21,67,68]. Following primary antibody incubation, slides were washed three times in PBS, then incubated 2 h with Alexafluor 488 (1:500) and Alexfluor 647 (1:500) in PBS + 10% NGS. After secondary antibody incubation, slides were washed in PBS three times for 10 min and then cover slipped with ProLong Gold with SYTOX-Deep red (ThermoFisher). After coverslipping, slides were allowed to dry at room temperature overnight, then edges were sealed with nail polish. Slides were stored at 4 °C. Patterns of GLU expressing neurons were further confirmed with vglut2 mRNA expression.

### Surgery and neurophysiology

Although the size of males used for physiology experiments varied (11.4–18 cm), all males were in breeding condition. Surgical methods for exposing the brain and occipital nerve roots and then recording the vocal motor volley, or fictive call from one of the roots, follow those from prior studies[45,46]. Briefly, fish were deeply anaesthetised during all surgical procedures by immersion in aquarium water containing 0.025% benzocaine (ethyl p-amino benzoate; Sigma-Aldrich, St. Louis, Mo., USA). To facilitate precise targeting of midbrain vocal sites, blood vessels in the optic tectum were cauterised and the dorsal-most portion of the tectum was resected. Following surgery, fish received an intramuscular injection of bupivacaine anaesthetic (0.25%, Abbott laboratories, Chicago, Ill., USA) with 0.01 mg/ml epinephrine (International Medication Systems, El Monte, Calif., USA) near the wound site that was subsequently administered every 4 h until euthanasia along with an intramuscular trunk injection of pancuronium bromide (0.1–1 µg/g body weight), a muscle relaxant. Animals were next positioned in a plexiglass tank and gently perfused over the gills with artificial seawater at the same temperature as their home aquarium water (18–20 °C).

### Pharmacological manipulations

Solutions within fabricated glass micropipettes (~20 µm diameter) were pressure-ejected using a picospritzer (Biomedical Engineering; ~25 nl/ pulse, repeated at ~2 Hz, at 25–30 PSI)[15,23,45]. Fictive calls were monitored continuously before, during and after each injection. Injections of gabazine (Tocris Biosciences; 50-150 nl total volume, ~25 nl/pulse, 1 mM in 0.1 M PB) within forebrain vocal regions induced persistent fictive calling. To block such calling, we pressure injected either muscimol (Sigma-Aldrich, 10 mM in 0.1 M PB; n = 2) or a NBQX (Sigma-Aldrich, 2.5 mM in 0.1 M PB) and APV (Sigma-Aldrich, 2.5 mM in 0.1 M PB) cocktail (n = 2; ~250 nl) into the PAG during extended periods of fictive calling. Bilateral injections were performed sequentially with one micropipette. Either 5% Fluorescein or 5% Alexa-Fluor 568 (Invitrogen-Fisher Scientific) was included in the pipette solutions to facilitate locating injection sites. For pharmacological mapping of midbrain or hindbrain vocal sites, micropipettes were filled with either 0.5M L-glutamate (GLU, Sigma-Aldrich) in 0.1 M PB (pH8), or 1 mM

gabazine (Tocris Bioscience) in 0.1 M PB. To localise the injection sites, 5% neurobiotin was included in the pipette solutions. After baseline recordings searching for vocal sites that were identified using electrical stimulation (see[45]), a glass micropipette was guided to possible vocal sites using surface landmarks, effective depths for electrical stimulation, and previous mapping studies[15,23,45,69]. The pipette solution was pressure-ejected using the picospritzer (total volume, 25–100 nl of GLU or 100–300 nl of gabazine).

Each fish was injected unilaterally with either GLU or gabazine. For each site injected with GLU, injections were repeated at least three times, with at least 5 min between each injection if no calling was evoked, and at least 5 min after the cessation of calling if calling was evoked. Response to at least one injection of GLU was considered a 'positive' site. Typically, two specific sites were tested for GLU response in each animal, one on each side of the brain, with more than a 20 min interval between injections in distinct sites that allowed for complete recovery from any effect of the first injection. Midbrain vocal responses following gabazine injection occurred at longer latencies but persisted for longer durations. Because of this, longer periods between injection sites were used (at least 30 min without calling before a new site would be tried), and if the first site produced calling, only one site would be tested in that fish. For both drugs, sites that did not respond were only classified as non-vocal if the fish maintained fictive vocal responses to electrical stimulation throughout the experiment, and later histological examination showed that tracer dye had in fact been released into the brain.

The locations of midbrain injection sites were confirmed in 52 cases for GLU, and 18 cases for gabazine. 26 GLU and 9 gabazine cases were located within the designated boundaries of PAGcs and PAGcd (Fig. S8d, e), and thus considered hits; fictive vocal data from sites outside these regions occurred rarely, and only ever at long latencies. The locations of hindbrain injection sites in the vocal prepacemaker nucleus (VPP) were confirmed in 3 cases for GLU. Following transcardial perfusion with a cold teleost Ringer rinse, brains were removed, post-fixed for 24 h in 4% paraformaldehyde in 0.1 M PB, and then sectioned as described earlier. Where neurobiotin was used as a marker, it was processed for visualisation using either diamionobenzidine (DAB) or Strepavidin-AF596. Injection sites were checked using light or fluorescence microscopy (Nikon Eclipse E800), and the centre of the injection site was taken to be the point of densest stain, or at the end of a visible pipette track in the tissue. These points were registered to a common reference brain based on distance from local anatomical landmarks within the midbrain and hindbrain.

## Physiology data analysis
Latencies and durations of pharmacologically induced calling were measured using Clampfit 9 (Axon Instruments). Latency was measured from the mechanical artifact in the recorded trace created by the first injection pulse to the first spike in a fictive vocalisation. Duration was measured from the first spike in a fictive vocalisation to the last spike of the last fictive vocalisation. In some cases, gabazine injection initiated persistent calling, sometimes lasting up to ~1.5 h; calling was considered terminated when the experiment ended, and the recording ceased. Physiology data (axon binary files) were imported using a preexisting MathWorks script. For analysis of call rate (calls per min) data were then processed using Matlab (Mathworks) with custom written scripts (I.H.B.). For analysis of spontaneous and forebrain-gabazine induced calling rates (and their silencing by midbrain injections), we calculated the number of fictive calls per min for each min of the experiment, before, during and after attempted midbrain silencing.

## Processing and analysis of natural and fictive calls
Physiology data and sound recordings were processed using a custom-written script in MATLAB (N.A). A lowpass filter with a cutoff of 1 kHz was applied to all recordings followed by a moving average to smooth the recordings. Recording amplitude were normalised to the maximum value and peaks above 25% of the maximum value were detected using internal MATLAB functions. Temporal locations of peaks were used to extract call duration, pulse repetition rate (PRR) and intergrunt interval.

To investigate whether PAG evoked fictive calls (GLU or gabazine microinjections) mirrored natural calls, we performed UMAP analyses using the *smallvis* package in R. In these UMAP analyses, for both grunts and growls, we included six variables: call duration (sec), coefficient of variation (CV) for call amplitude (AMP), average PRR, CV for call PRR, percent change in AMP from the lowest peak to the highest peak, percent change in PRR from slowest gap between pulses to the fastest gap between pulses. Similar analyses were used to compare fictive and natural calls.

## Super-resolution microscopy
Three type I males (13.3 – 18.7 cm, standard length) received application of neurobiotin tracer (Vector, Burlingame, CA) to the vocal nerve that innervates the swim bladder musculature (see[37]). Postsurgery survival time was 5-7 days. IHC protocol followed the outline above with incubation of PSD95 (mouse monoclonal, Millipore, MAB1596; 1/500) and vglut1/2 (rabbit polyclonal, synaptic systems: AB135 503; 1/250) at room temperature overnight. The next day slides were washed twice in PBS for 10 min and then incubated in secondary antibodies (anti-mouse-567 and anti-rabbit 647; Molecular Probes; 1/200). At this time, we also added a fluorescent-conjugate streptavidin (Alexa 488; Molecular Probes; 1/200). After secondary antibody incubation, slides were washed in PBS three times for 10 min and then cover slipped with ProLong Gold with SYTOX-Deep Red (ThermoFisher). After cover slipping, slides were allowed to dry at room temperature overnight, then edges were sealed with nail polish. Slides were stored at 4 °C.

Multicolour images using SR-SIM were acquired with a Zeiss Elyra S.1 SIM system using a 63×/1.4 oil immersion lens and ZEN 2012 software (Zeiss). Z-stacks (0.1 μm step size, 3 μm thick) were acquired using five grid rotations and five phases at each plane. To maximise signal-to-noise, images were acquired within the first 5 μm from the tissue/coverglass interface. SR-SIM images were processed in ZEN, followed by generation of maximum projections and single optical plane images with ImageJ.

## Image acquisition
Except for SR-SIM, all images were acquired on a Zeiss 510 inverted confocal microscope using 20x air objective and Zen software (2009 version 6.0, Zeiss). For each brain region, we acquired z stacks at 2 μm intervals throughout the entire distance of the brain region of interest. Maximum intensity projections of these z stacks were created using ImageJ. For high resolution inserts for the catFISH experiments, we used a 25x water objective with 1.5x digital zoom to acquire z stacks at 1 μm intervals throughout the PAGcs or PAGcd.

## Image analysis
Cell counts were generated by two observers blind to the behavioural condition of the animals. For each brain region, we performed cell counts on 2-3 adjacent sections. Measures of *c-fos* mRNA were performed on max intensity projections of confocal images in imageJ. To further validate our counts, we used *Cellpose v2*, an automated machine learning cell counting package[70]. We found that minimal training of the base 'cyto' model was sufficient to yield results that were strongly correlated with human observers ($r^2 = 0.78$; between human observers $r^2 = 0.89$). Specifically, we trained this base model using a human-in-the-loop approach, on 6 PAGcs images (1758 individual PAGcs neurons with *c-fos* mRNA label). During this training, a human observer manually removed cells that only had nuclear signal. To obtain a cell count for each region, we averaged cell counts from each blind observer and the cell counts generated by the Cellpose model.

To assess how *c-fos* label overlapped with GABA and GLU label, an observer blind to the condition of the animal performed counts on z-stacks of confocal images. This allowed the observer to clearly see when *c-fos* expression appeared in a neuron (Fig. S3).

Next, to investigate whether PAG neurons were active in one or both vocal behaviours, an observer blind to the condition of the animals first counted the number of intron expressing cells (nuclear restricted; red signal) by scanning through a z-stack (2 μm steps) to determine the appearance of intronic signal that overlapped with SYTOX (nuclear stain; blue). These counts were used to determine the total number of intronic signal in each zone. The observer next determined whether each cell with intronic expression also had cytoplasmic label (reactivated cells; activated in both behaviours 1 & 2). These two values were used to determine the percent of nuclear labelled *c-fos* cells with cytoplasmic *c-fos* mRNA expression (reactivation index).

### Statistical analyses

All analyses were performed in R (v3.3.2). A summary of all statistical tests can be found in table S2. Data were [log (1 + x)] transformed to achieve normality, as Q–Q plots and Shapiro-Wilk tests indicated that these transformations yielded more normally distributed data. The analyses where we did not perform this transformation were the cat-FISH reactivation index data, as these data were percentages. In all analyses, significant main and interaction effects were further examined with *post-hoc* comparisons, using Benjamini-Hochberg (BH) corrections to account for multiple contrasts. All analyses are summarised in detail in Table S2.

We ran a one-way analysis of variance (ANOVA) for each of the brain areas of interest to determine whether the number of *c-fos* mRNA-expressing cells significantly differed between the three behavioural groups. Linear regressions were used to determine whether *c-fos* mRNA expression in PAGcs or PAGcd correlated with the amount of courtship or agonistic calls that an animal produced.

Several analyses were used to investigate the catFISH data. First, to determine the time course of *c-fos* intronic and coding expression, we used independent sample t-tests expression differences between agonistic-silent and silent-agonistic. Next, ANOVAs were used to analyse overlap between courtship and agonistic neurons in the caudal PAG. An independent sample t-test was used to investigate whether vocal neurons were also active during feeding. Linear regressions were used to determine whether *c-fos* intronic expression in PAGcs or PAGcd correlated with the amount of agonistic calls that an animal produced. We also used regressions to investigate how intronic expression was related to the amount of swimming during the second behavioural period in catFISH agonistic trials (see above).

An independent sample t-test determined whether foraging neurons were distinct from agonistic neurons. Independent sample t-tests were used to investigate the latency and total duration of fictive calls following microinjection of GLU or gabazine into the caudal PAG. To analyse acoustic differences (call duration, CV PRR, and CV AMP) between the PAGcs and PAGcd for both GLU and gabazine experiments, we ran linear mixed models. In these models, animal identity was used as a random effect. Similar analyses were used to investigate the effects of GLU in VPP compared to natural calls.

For all ANOVAs and LMMs eta squared (η2) effect sizes were computed using the 'sjstats' package. For all η2 analyses, values above 0.25 were considered robust and meaningful differences between groups[71]. In addition, we calculated Cohen's d effect sizes for all for all independent sample *t*-tests comparisons[71].

### Reporting summary

Further information on research design is available in the Nature Portfolio Reporting Summary linked to this article.

## Data availability

All data can be found in the supplementary "source data file". Source data are provided with this paper.

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

## Acknowledgements

The authors thank Bruce Land, Kirstin Hagelskjaer Petersen, Jiayu Dong, Hongshu Ye and Lu Lu for the 3-D rigid model of a midshipman fish male; Margaret Nelson for animal drawings; Dipayan Chaudhuri for tract-tracing studies of optic tectum and cerebellum; Boris Chagnaud, Ni Y. Feng, Jesse Goldberg and Michael Brainard and members of his lab for discussion or comments on earlier versions of the manuscript; Joe Fetcho for the generous use and guidance of his Zeiss LSM510. Sound recording for monkey in Fig. 1 courtesy of U. Jurgens. Research support from NSF IOS1656664 (A.H.B.) and Cornell Neurotech Mong Fellowship (N.A.).

## Author contributions

Conceptualisation: E.R.S., I.B and A.H.B. Methodology: E.R.S., I.B., N.A., J.T.P. and A.H.B. Investigation: E.R.S., I.B., N.A., W.F., J.T.P., C.H.R., M.A.M. and A.H.B. Visualisation: E.R.S., I.B., N.A. and J.T.P. Supervision: A.H.B. Writing—original draft: E.R.S., I.B. and A.H.B. Writing—review & editing: E.R.S., I.B., N.A., J.T.P. and A.H.B.

## Competing interests

The authors declare no competing interests.
