## [Peer Review File · Nature Communications]

Midbrain node for context-specific vocalization in fishReviewers' comments:

Reviewer #1 (Remarks to the Author):

This is a well written manuscript from the laboratory that has a great deal of experience with this model system. The strength of the paper lies in its level of details and multidisciplinary approach. The authors show strong evidence the fish PAG circuit is comparable to mammals and this pathway has been conserved through evolution. Very few fish vocalizations networks have been extensively studied as the midshipman and the lab's previous studies formed a strong foundation on which to conduct and support their current experiments.

I have a few suggestions for the authors

Fish vocalization experiments are challenging and I applaud the authors for being able to evoke different types of vocalizations. However, the calling duration for each "behavioral" periods was not well described (i.e. up to five min). Did the fish call for exactly 5 min? Where there any pauses. Could call duration and hence vocal motor activity be correlated with the neuronal activity? The intercall interval was very variable (25 to 35 min) and how may this have influenced the data. Also, during the agonistic behaviors, the grunts should have been fairly easy to quantify. How did total number of grunts correlate with the brain activity. Quantifying vocal output with neural activation, could strengthen the manuscript

Unfortunately, this may not be possible because of the relatively low sample sizes. However, the sample sizes are inconsistently noted throughout the manuscript and are interspersed through the text (methods N=3 given for super resolution microscopy), figures and figure legends making it very hard to follow how many fish were examined for each task. This also leads to some concerns about the statistical power to obtain the conclusions. There are also non parametric (Fig 2, 5) and parametric (Fig 4) analysis that needs to be clarified in the methods. For example, were the percentages in Fig 4 f transformed prior to analysis. In general, the entire statistical analysis section could be strengthened by more detail. Also, more consistency with sample sizes in the text and figures would enable the experiments to be more easily followed. This is a compelling manuscript and further attention and organization of the statistical analysis would strengthen the paper.

Both the agonistic and foraging activity involved considerable movement while the courtship activity did not. How might the locomotion impacted the brain areas? The foraging data is the weakest part of the paper. This fish is an excellent model for vocal communication but is not well known for locomotory behaviors. It was unexpected that nest guarding and vocalization males even consumed prey although perhaps eating cessation occurs after eggs are present. Without further detail, I would suggest omitting the foraging experiments as there could be quite a bit of variability. They are not quantified, they suffer from small sample size and do not add to the manuscript.

Line 153 Missing a word?

Line 343 This seems like an apple/orange discussion. Prey capture vs escape behavior and may need further clarification to its relevance

Line 363. The grunt may be the basic unit of toadfish vocalization. However, given the widely varying nomenclature and lack of any quantitative assessment of how to classify vocalizations in the fish literature, extending this statement to "many" teleost species may be too broad and is not supported

Line 445 -When the terracotta lid was removed, did the males float out of the nest?

453 Prey capture can be highly variable. This needs further details to warrant inclusion

Figures. The micrographs are very good. However, some figures (6 through 8) are overly busy and are difficult to read. The authors may want to streamline these figures and move some of the information to supplemental materials especially the pie charts

There is no indication of what the box and whisker plots are showing (are confidence intervals 10/90, or 5/95) and the points are presumably outliers?

Figure 2H Very difficult to see the fish ; J through L. Difficult to see label. Maybe show an enlarged inset?

Figure 5 – No sample sizes reported

Figure 6 seems overly busy and could be divided into two figures. At least e,f, g and h could be better labelled.

Reviewer #2 (Remarks to the Author):

Increasing evidence suggests that the midbrain periaqueductal gray (PAG), especially the dorsal portion in mammals, is a conserved brain center critical for orchestrating the basic action plan for the execution of instinctive survival behaviors that include escape, attack, parenting and breathing. Many studies suggest the PAG is remarkably conserved across phyla and that it also acts as a critical output node for the production of vocal behaviors. Most of what we know about the functional role of PAG derives from work in tetrapods, with most studies performed in mammals, and arguments that PAG acts a conserved structure in the production of innate survival behaviors are based primarily on hodological arguments of how it is connected to pathways necessary for the production of these innate behaviors. Careful physiological functional studies have not been performed in non-tetrapod species.

In a physiological tour-de-force, Schuppe et al. perform a series of careful experiments to show that PAG is both necessary and sufficient for the production of the complex vocalizations that midshipman fish produce. This study therefore provides important new evidence that the PAG, at least in the context of the production of vocal behaviors, acts as a conserved motor output node for the orchestration of these complex innate behaviors. In addition to this important evolutionary contribution, the authors also take full advantage of the behavioral repertoire of these fish to show that the PAG, but not its downstream hindbrain targets, encode two distinct types of vocal behaviors, a behavioral dichotomy that can't be easily exploited in other animal models. Specifically, using fluorescent *In Situ* hybridization of the immediate early gene *c-fos* they are able to show, using a well-balanced experimental design coupled with differential cellular localization of nuclear vs. cytoplasmic transcripts, that neurons in portions of the PAG, specifically the caudal superficial portion (PAGcs), show differential activation of subsets of neurons for these two different vocal behaviors, a pattern that is not observed in the target vocal prepacemaker nucleus (VPP). This finding implies that unique subsets of neurons in this structure are activated to produce these different vocal behaviors. Finally, the authors show convincingly that stimulation of PAGcs, either by direct activation of glutamatergic neurons or by disinhibition of these same areas evokes vocalizations that have many of the characteristics of natural calls with latencies being significantly longer when vocalizations are evoked by disinhibition, results that mirror those obtained in mammals.

Primary comments:

1. The authors show clear differential activation in PAG for the different context-dependent vocalizations. This is implicitly interpreted as being caused by motor-based activation of these different neuronal subsets. However, these findings could potentially also be explained by differential auditory activation given that the PAG is known to be highly integrative. Ideally, it would be good to provide a control where IEG expression is measured in de-vocalized fish (probably the best control). It is possible that fish will not vocalize under such conditions or that such experiments are not feasible. If that is the case, the authors should certainly consider discussing the role that auditory inputs might play in driving *c-fos* expression in these parts of PAG. It is even possible that the authors might well have information (maybe PAG does not receive auditory inputs in midshipman fish?) that address this concern and make it a moot point.
2. One of the hallmarks of the PAG, is that it is suggested to contain areas that are differentially activated by different instinctive behaviors. Consistent with this idea, the authors show that different vocal behaviors activate overlapping but nevertheless distinct subsets of neurons in the PAG. The authors also perform an experiment to compare neuronal activation patterns between agonistic vocalizations and a very different behavior, in this case foraging behavior. Here (unless I missed it) they only look at foraging specific *c-fos* expression in PAGcs and PAGcd to compare activation evoked by calling behavior. Consistent with the idea that PAG drives different types of behavior, only a small subset of neurons is activated by both behaviors. While this is interesting, I would have liked to see analysis of other parts of what is defined as PAG in the fish to show that there is actually an area that is activated primarily by foraging but not vocalization.
3. A large portion of the study is devoted to the use of neurotransmitters or pharmacological agents to initiate fictive vocalizations. While the findings are convincing, it would be useful to provide more information regarding how well injections were restricted to the intended target as well as the inclusion of controls to show that injections of glutamate, say in areas activated by foraging, do not produce vocal behaviors.

Minor comments:

Line 42. No references are shown so it is unclear what the authors mean by a circuit of 100s of neurons that can both initiate and patterns complex vocalizations. Do these circuits really only contain 100s of neurons?

Line 55 and Figure 1. The statement in the text does not match very well what is represented in the figure given that it does not contain any information about organizational pattern for the limbic stream.

Line 64 – please define the term “sensu” which does seem to be defined in the suggested reference.

Figure 2 – j through o would benefit from a bit more description in the legend regarding the staining pattern shown in the figure.

Figure 3 – it might be good to have a panel showing overlaying the biotin injection site with a c-fos staining of the VPP during vocalization.

Line 153 – typo “...calls [had] high amounts ...”

Line 170 – It would be helpful to get a better sense of the actual number of neurons showing c-fos staining in the PAGcs and PAGcd for both foraging and vocalizing because the figures only show percentages. Also the figure legend describing l through n is difficult to follow. Please consider revising.

Line 192 – this is just a suggestion. But it might be nice to introduce this section by stating that behaviors could be initiated either by direct activation of by disinhibition and then perhaps provide examples of how these different strategies are used in other systems or contexts.

It might be worth reminding the reader how to think of the VPP and what would likely be its analogue or homologue in mammals and birds.

Line 324 (and in several other places) – Fig 8 is written as Figure 80

Reviewer: Marc F Schmidt

Reviewer #3 (Remarks to the Author):

In this paper, Schuppe and colleagues analyze activity in the periaqueductal gray (PAG) nucleus of the brains of a type of teleost fish known as toadfish, during two types of vocalization events (courtship humming calling and defensive grunts/growls). The authors use c-fos staining to claim that these different types of vocalization episodes are correlated with different spatial patterns of c-fos activity in the PAG. They then perform tracing studies to claim that the zones they delimited in the PAG have connections with a hindbrain site known as the vocal prepacemaker nucleus (VPP), which previous studies have shown to be part of the vocalization centers in the fish brain. Next, they perform double in situ hybridization with temporal and spatial resolution (catFISH technique) to probe the responsivity of each PAG neuron, concluding that most cells are responsive to either one or the other type of vocalization situation (courtship humming calling or defensive grunts/growls). After conducting staining experiments that claim that most c-fos responsive cells are glutamatergic, the authors conduct a series of pharmacological manipulation experiments with glutamate receptor antagonists or glutamate injections to probe the necessity and sufficiency of the PAG for both types of vocalization outputs.

The experiments in this manuscript lack important controls, were not properly interpreted or are not convincing, and therefore I do not feel that the conclusions drawn from the data are supported.

Major points:

1) Several hypotheses in the paper are either presented as evidence or simply not supported by the data. For example, the observation of c-fos staining in the PAG during episodes of vocalization is certainly much distant from the amount of evidence needed to claim that the nucleus is a "vocalization center", as seems to be claimed by the authors throughout the study. An example of this is found in line 113: "These results conclusively demonstrate active involvement of lateral PAG neurons in vocal control in toadfishes"; this is an overstatement, as one would not be able to conclude that a nucleus is involved in controlling a certain motor behavior based solely on c-fos staining patterns.

2) The separation of PAG into PAGr, PAGcd and PAGcs "zones" based on c-fos staining is not convincing. Activity in these areas is not spatially coherent and there is no neuroanatomical indication that each constitutes a functional unit. Other studies even considered PAGcd to be a separate nucleus. Nomenclature issues apart, the c-fos stainings are not convincing in delimiting coherent patterns of activity to justify this zonal categorization, as the somata are diffusively distributed in the multilayered structure that the authors claim to be the PAGcd.

3) Importantly, Fig. 2i shows that there is no increase in the numbers of c-fos+ cells during courtship as compared to controls (overall statistical significance in the ANOVA test is not enough). Another problem is that the image in Fig. 2n is not representative, as the graph in panel i does not indicate any statistically significant difference compared to the 'silent' controls, but the image shown does display a clear difference.

4) It is also hard to judge if the c-fos cell counting was properly done, as the low-mag images in Fig. 2j-o do not make it clear that the green 'dots' dispersed throughout the image are real green-stained c-fos+ cellular nuclei. I see at least 10 green specs that do not coincide with the underlying blue SYTOX nuclear staining.

5) The tracing experiments presented in Fig. 3 are hard to interpret. Based on evidence from previous publications that the hindbrain VPP nucleus controls motor activity necessary for vocalization, the authors decided to conduct retrograde tracing to show that the VPP receives inputs from the PAG. Although the staining shown in the PAG is evident, the separation between the staining in the different "zones" defined by the authors (PAGs, PAGcd and PAGcs) is neither evident nor properly quantified. Important controls are also missing, such as showing that the injection site is restricted to the VPP or that other known VPP inputs are also labeled. Importantly, detecting retrograde labeling from the VPP in the PAG is not enough to classify the latter as part of a vocal circuit or as a "vocal hotspot", as claimed by the authors in line 126 (moreover, the word "hotspot" is most certainly meaningless jargon in this sentence).

6) The heterogeneity of cellular responses (as judged by c-fos expression) during different types of vocalization events was investigated using the catFISH procedure (Fig. 4). This experiment would be interesting to define the map of activity concomitant with (not "in response to", as argued by the authors) vocalization events, but the results are far from convincing:

6.1 - The critical controls in Suppl. Fig. 1 are simply not good enough: (a) I see clear green staining in Suppl. Fig. 1b (Silent-Agonistic situation), in which case one would expect to see only red staining related to the c-fos intronic probe; (b) the red staining related to the c-fos intronic probe is not evident anywhere in Suppl. Fig. 1 or in Fig. 4; there should be very clear foci (preferably two per nucleus) inside the blue nuclei in the picture, and I do not see any; the images in Suppl. Fig. 2e,g are better but still not clear enough since all fluorescent channels are merged; (c) likewise, the green staining in Suppl. Fig. 1c (Agonistic-Silent situation) should be predominant, but I cannot judge to which extent red staining is also present, since only the merged image is presented and some red staining is clearly apparent in some cells; importantly, the green staining should be both nuclear and cytoplasmic, as it is the result of staining with a c-fos probe designed against the coding region, but I cannot judge if the blurry (and sometimes punctate) pattern of green staining is truly cytoplasmic in these low-mag images; (d) finally, no quantification is presented for these critical control experiments in Suppl. Fig. 1.

6.2 - Most significantly, the same criticism goes for the whole of Fig. 4: no evident red staining is seen anywhere in these low-mag merged images, and the green staining is either blurry punctate (and apparently not coinciding with any blue nucleus) or blurry and not convincing as cytoplasmic staining.

6.3 - Moreover, the graphs in Fig. 4f claim that consecutive episodes of vocalization of the same type (Courtship-courtship, for example) lead to largely overlapping staining with the c-fos coding and intronic probes, as should be the case if the catFISH procedure worked correctly, but I cannot see such significant level of overlap in the corresponding image in panel g. Likewise, the agonistic-agonistic situation should result in largely overlapping subpopulations of c-fos+ cells according to the graph in panel f, but I cannot see that in the image in panel h. Without such controls, the

interpretation of panel i, which contains the critical comparison between the “agonistic” and the “courtship” situations, is rendered impossible.

6.4 - I also think it is very confusing to represent the bar graphs in Fig. 4 using ‘green’, ‘red’ and ‘yellow’ colors, as these colors might give the impression that the ‘yellow’ bars refer to concomitant staining with both c-fos coding (green) + c-fos intronic (red) probes, which is certainly not the case.

6.5 - Finally, the numbers of replicates in each comparison are not properly described in the figure legend (does n=5 mean 5 fishes or 5 sections from a smaller number of fish?).

7) Even considering that the analyses and quantification in this catFISH experiment are sound, the conclusions that there is little overlap between the c-fos staining for the agonistic versus the courtship vocalizations and that such small overlap implies that both situations activate largely distinct subpopulations of PAG cells are not supported by the data. The graph in Fig. 4f shows close to 50% overlap in both PAG regions analyzed, so the conclusion that the subpopulations of active PAG cells are disjunct is not warranted. And in the PAGcs, which later experiments in the paper interpreted to be the primary output to the vocal hindbrain centers to initiate vocalization, the level of overlap of c-fos staining for both types of vocalization events is close to 70%.

8) Irrespective of these technical limitations and the poor quality of staining, microscopy images, and quantification, the authors conclude that there is little overlap between the c-fos staining in PAG neurons during vocalization and during a non-vocalizing activity (foraging) in Fig. 4m,n and that this result implies there is “little evidence of functionality of the PAG zones outside of their role in calling” (line 179). I cannot objectively understand this statement and I do not see how the analysis of c-fos activity in two brain regions when the animal is performing only two types of tasks can lead to any conclusion about functionality, of either the circuit or the specific nuclei analyzed.

9) The characterization of c-Fos+ cells in the PAG relative to their glutamatergic or GABAergic phenotypes is based on unconvincing staining as well (Fig. 5):

9.1 - The staining in Fig. 5d-j used antibodies against glutamate and GABA, which is far from ideal, as the resulting fluorescent patterns are diffuse, difficult to correlate with specific cell bodies and with high non-specific background staining.

9.2 - I see very clear overlapping staining for glutamate and GABA in a number of instances, such as in the insets in Fig. 5e,f and in Suppl. Fig. 3, which is hard to understand and reconcile with the conclusions that the vocalization-related c-fos activity is found mostly in glutamatergic cells.

9.3 - Moreover, the c-fos staining in Fig. 5 is not good, as it is a mixture of punctate staining, background staining, and some staining that appears to occupy the whole cell body. Again, it is hard to correlate the staining with the underlying nuclear staining, as all images are low-mag, presented as merged fluorescent images, and no SYTOX staining was performed or shown.

9.4 - Importantly, the method the authors used for quantifying the levels of staining for either glutamate or GABA in Fig. 5a,b is not ideal: numbers of c-fos+ cells that co-stain for GLU or GABA do not mean much, as the authors themselves state that the overall numbers of “cells” stained with GLU or GABA differ dramatically in all PAG zones. Therefore, the best strategy would be to quantify the percentages of c-Fos+ cells that are either GLU or GABA-positive, a procedure that seems to have been performed and shown in Fig. 5d, in pie graphs that are neither quantitative, nor statistically explored, nor properly explained (scale?).

9.5 - These experiments must preferably be conducted with antibodies against glutamate or GABA vesicle markers (vGLUT, vGAT, or GAD) or preferably still with in situ hybridization probes against the respective genes, as these staining patterns would be very clearly associated with cell bodies, allowing the precise quantification of an overlap with the c-Fos staining pattern.

9.6 - The sentences in line 195 discuss that the VPP receives inputs from the PAG and that glutamatergic signaling in the VPP is important for vocalization, but no experiments were performed to show that glutamatergic projections from the PAG vocalization responsive cells reach the VPP.

9.7 - In sum, I am not convinced, based on the data and graphs presented, that “activation of GLU-, but not GABA, containing PAGc neurons are essential for initiating social context-specific vocal output”, as the authors claim in line 196. Even if the calculations were correct, why would the quantification of c-fos/GLU/GABA co-staining indicate anything about the essentiality of a nucleus in initiating a behavioral output?

10) In the experiments to pharmacologically manipulate the activity of PAG neurons (with either GABA agonist or glutamate receptor antagonists) in Fig. 6, because the animals are anesthetized the authors used a GABA receptor antagonist injection to elicit patterns of vocalization measurable by evaluating the occipital nerve, instead of using the more natural situation in which the animals vocalize either to the dummy fish or to attract females in the basal humming vocal events. Even

though some effort is undertaken to compare the features of pharmacologically-evoked calls and the natural responses using dimensionality reduction algorithms (UMAP) in Fig. 6, the results do not seem to show that these responses are similar, as claimed by the authors: for example, I can clearly see a difference between the responses of the natural grunts (in green in Fig. 6f) and the grunts induced by injection of glutamate into the PAGcs (in red).

11) Moreover, the authors do not show the images from control experiments to evaluate that the glutamate injections into the PAG target solely this nucleus and, importantly, just one of the PAG zones analyzed. Since the PAGcd and PAGcs responses are being contrasted, these controls are critical. Given the leakiness and potential variability associated with stereotaxic pharmacological injections into the brains of a non-standard species, I consider that these assays are prone to the vagaries of experimentation and I am not sure that the data shown are enough to assess such variability in order to allow meaningful conclusions to be drawn about the sufficiency of the PAG and its "zones" in eliciting various types of vocalization patterns, as claimed by the authors.

12) All supporting data must be shown. 'Data not shown' statements are not sufficient and are not allowed as per Nature's reporting guidelines.

13) Given these technical and interpretational problems, I am not convinced by the pharmacological manipulation experiments and therefore cannot support a publication stating that the PAG is both necessary and sufficient to drive different types of vocalization patterns in the model fish used (line 256).

Minor points:

1) There is no need to keep Figure 3 as separate, given the minimally important set of data presented therein for the whole story. Either join these data to another figure or transfer them to the supplementary material.

2) Fig. 1 and the first panel in Fig. 2 should not be presented before the main results. They are prior data and could be transferred either to the supplementary material or simply explained in the text.

3) The "2" bottom-left inset in Fig. 4n does not correspond to the indicated region in the low-mag image.

4) The meaning of letters 'a', 'b', etc. above bars in bar graphs after ANOVA should be described in the legend, as being the result of post-hoc pairwise comparisons.

5) Fig. 1 legend is cut short.

6) Panels in Fig. 2j-o are misaligned.

7) The Methods section should be carefully revised, as the text is very confusing and inaccurate in a lot of instances. For example, in the sentences starting in line 504, the authors refer to a "biotin blocking reagent" (without description) and later they refer to a "streptavidin blocking reagent" and to a "blocking buffer". Each solution in the experiments must be clearly specified (concentrations and reagent manufacturers) and the methods should be described using a nomenclature system that is both unique and standard.

8) In line 522, the description of catFISH is confusing. If FITC and DIG labeled probes were used, why have the authors labeled the FITC probe with Cy3? If this is not the case, then the text is confusing and must be rewritten. I imagine that they used an HRP-conjugated anti-DIG antibody, followed by Cy3-tyramide labeling, for the c-fos intron probe (red fluorescence in the microscopy images), but this is not what the text describes. Moreover, the brands and part numbers of all critical reagents must be listed throughout the main text.

Manuscript title: At the outset, we note that we have changed the original title of the manuscript, “A conserved midbrain circuit motif for context-specific vocalization in fish”, to the following: “Midbrain node for context-specific vocalization in fish”. This was done, in part, in response to a comment in Review 3 regarding that some of our findings may be “overstated”. Given that the level of detail presented in our study only compares to studies in mammals, we thought it prudent not to use the term “motif”, which would imply the existence of the PAG node reported here as present in other vertebrate clades; this awaits further study.

Responses to Reviewers' comments (except where indicated, all references cited appear in the reference list of the revised main text of the manuscript):

Reviewer #1 (Remarks to the Author):

This is a well written manuscript from the laboratory that has a great deal of experience with this model system. The strength of the paper lies in its level of details and multidisciplinary approach. The authors show strong evidence the fish PAG circuit is comparable to mammals and this pathway has been conserved through evolution. Very few fish vocalizations networks have been extensively studied as the midshipman and the lab's previous studies formed a strong foundation on which to conduct and support their current experiments. I have a few suggestions for the authors

- 1. Fish vocalization experiments are challenging and I applaud the authors for being able to evoke different types of vocalizations.**
 - a. However, the calling duration for each “behavioral” periods was not well described (i.e. up to five min). Did the fish call for exactly 5 min?**

Response: All fish in the agonistic catFISH experiment produced agonistic calls throughout each five-minute period. However, there was variation in the total number of agonistic calls produced across animals (range: 2-21 grunts; range: 0-16 growls). We have added these details to the Methods (see end of section titled “Agonistic (grunt, growl) vocalization”).
 - b. Where there any pauses. Could call duration and hence vocal motor activity be correlated with the neuronal activity? Also, during the agonistic behaviors, the grunts should have been fairly easy to quantify. How did total number of grunts correlate with the brain activity. Quantifying vocal output with neural activation, could strengthen the manuscript. Unfortunately, this may not be possible because of the relatively low sample sizes.**

Response: We have performed new analyses to investigate how vocal behavior is associated with *c-fos* mRNA and intronic expression. We found that (1) animals that produced more courtship hums had a greater number of *c-fos*⁺ cells in both the PAGcs and the PAGcd (Fig. 2j, l) and (2) animals that produced more agonistic calls had a greater number of *c-fos*⁺ cells in the PAGcs, but not in the PAGcd (Fig. 2k, m). In sharp contrast, the PAGrs exhibited no relationship between *c-fos*⁺ cell number and the number of courtship or agonistic calls (Supplementary Fig. S3d, e). Together, these findings suggest that the amount of activation in either caudal PAG zone is related to the number of agonistic and/or courtship calls produced. Similar findings have been reported in both birds¹ and mice².
 - c. The intercall interval was very variable (25 to 35 min) and how may this have influenced the data.**

Response: Inter-call intervals varied in this way only when both calling periods involved

courtship calling. We have now added correlation showing that variation in this pause did not influence the amount of *c-fos* mRNA expressing cells in PAGcs or PAGcd ($p > 0.2$) (see end of paragraph 1, Methods section titled: "Behaviour experiments, courtship vocalization"). This suggests that the variation between courtship bout 1 (cytoplasmic expression) and the more recent courtship bout 2 (intronic expression) did not influence the findings.

2. **However, the sample sizes are inconsistently noted throughout the manuscript and are interspersed through the text (methods N=3 given for super resolution microscopy), figures and figure legends making it very hard to follow how many fish were examined for each task.**

Response: This was an oversight in the previous draft of the manuscript. In the revised manuscript, we state all the sample sizes in the relevant Method sections and the figure legends.

3. **This also leads to some concerns about the statistical power to obtain the conclusions. There are also non parametric (Ffig 2, 5) and parametric (Fig 4) analysis that needs to be clarified in the methods. For example, were the percentages in Fig 4 transformed prior to analysis. In general, the entire statistical analysis section could be strengthened by more detail. Also, more consistency with sample sizes in the text and figures would enable the experiments to be more easily followed. This is a compelling manuscript and further attention and organization of the statistical analysis would strengthen the paper.**

Response: We have now included a table (S2) in the Supplementary Materials that details each statistical test, the test statistic, degrees of freedom, p value, and effect size. Inclusion of the effect size (eta squared, Cohen's d, etc) measures for each statistic allows us to state, using established statistical guidelines, that each of our findings are very robust. We also add greater detail to our 'statistical analysis' section of the Methods, including the rationale for specific statistical tests.

4. **Both the agonistic and foraging activity involved considerable movement while the courtship activity did not. How might the locomotion impacted the brain areas? The foraging data is the weakest part of the paper.**

Response: Fortunately, we made videos of each 5 min behavioral period of agonistic calling during the catFISH trials. Since *c-fos* intronic expression is related to vocalizations produced in the last 5 min behavioural period of a trial, and its expression is correlated with agonistic vocalizations, we restricted our analysis of movement, namely bouts of swimming, to this time point (see "foraging" section of methods details). Video analysis showed that the time spent swimming during agonistic calling was not correlated with *c-fos* intron expression in either the PAGcs or PAGcd (see Supplementary Fig. S5i, j).

5. **This fish is an excellent model for vocal communication but is not well known for locomotory behaviors. It was unexpected that nest guarding and vocalization males even consumed prey although perhaps eating cessation occurs after eggs are present. Without further detail, I would suggest omitting the foraging experiments as there could be quite a bit of variability. They are not quantified, they suffer from small sample size and do not add to the manuscript.**

Response: We would prefer to keep these experiments in the manuscript as they show that PAG neurons active during vocalization are not associated with an unrelated behavior. We have observed nesting males feeding at nighttime. Field studies of nest-guarding males are consistent with these observations; we now cite supportive findings from two such

publications showing that nesting males gain weight and feed (DeMartini, 1988; Sisneros et al., 2009)^{3,4}. We fed males what are known as feeder goldfish (available in pet stores; included in our approved animal protocol) about once a week before behavioural experiments. In fact, foraging is one of the few behaviours we could reliably evoke without also evoking vocalizations. For all experiments, we both visually confirmed that animals foraged and examined stomach contents to confirm that they fully ate each goldfish provided to them. All of these details have been added to the Methods– see section titled “Foraging” under Behaviour Experiments.

As regards sample numbers in general, the fish used for our study are from a wild population that is accessible from their natural habitat only during the breeding season. We try to minimize the number that we capture to preserve the wild population. It is worth noting that previous studies that have run catFISH experiments have comparable (n=5-6; Ishii et al., 2018⁵), and occasionally lower (n=2; Wu et al., 2014) sample sizes. Furthermore, a new analysis of our time course controls (Supplementary Fig. S5a-f) exhibits similar trends to the original catFISH control experiments in Lin et al., which did not include replicates (Nature, 2011)⁶.

Line 153 Missing a word?

Response: Corrected in revised paragraph.

Line 343 This seems like an apple/orange discussion. Prey capture vs escape behavior and may need further clarification to its relevance

Response: We now refer to this behaviour as foraging. We have standardized this throughout the text and all figures.

Line 363. The grunt may be the basic unit of toadfish vocalization. However, given the widely varying nomenclature and lack of any quantitative assessment of how to classify vocalizations in the fish literature, extending this statement to “many” teleost species may be too broad and is not supported

Response: We agree and have revised this to refer specifically to toadfish.

Line 445 -When the terracotta lid was removed, did the males float out of the nest?

Response: When the lid was removed, males would remain buoyant for a short period of time. By the time the agonistic trials began, the fish no longer had the swimbladder fully inflated. To ensure that this did not affect behaviour we performed a pilot study showing that lid removal did not alter the total number of agonistic calls that males produced (n=6 males with no lid removed; 4 with lid removed). Importantly, other ways of disrupting humming (e.g., putting an object in the tank or tapping the top of their nest) would occasionally cause them to produce grunts in response to these disturbances. Finally, courtship-agonistic and agonistic-agonistic calling sequences had similar patterns of *c-fos* intron expression (Fig. 3f). We have added these details to the revised Methods in the section titled “Agonistic (grunt, growl) vocalization”.

453 Prey capture can be highly variable. This needs further details to warrant inclusion

Response: We ensured that all males ate the same number of fish in each trial. For example, fish in the foraging-foraging condition ate one fish during each 5 min behavioural period (1 and 2). Conversely, fish in the foraging-agonistic sequence ate only one fish in behavioural period 1. We both visually observed them eating and later confirmed consumption by inspecting stomach contents (also see response to earlier comment 5). We have added these details to the revised Methods section (see “Foraging”) .

Figures. The micrographs are very good. However, some figures (6 through 8) are overly busy and are difficult to read. The authors may want to streamline these figures and move some of the information to supplemental materials especially the pie charts

There is no indication of what the box and whisker plots are showing (are confidence intervals 10/90, or 5/95) and the points are presumably outliers?

Response: We have removed the pie charts and box plots from the main text figures. These panels are now in Supplementary Fig. S9. We also provide additional details that the box plots show 5 and 95% confidence intervals.

Figure 2H Very difficult to see the fish ; J through L. Difficult to see label. Maybe show an enlarged inset?

Response: We have added an insert to Fig. 2h showing a front view of a male midshipman in the nest. We have also added higher magnification views of label in PAGcs and PAGcd in bottom row of Fig. 2n-p, and of the PAGrs as inserts to Supplementary Fig. S3a-c.

Figure 5 – No sample sizes reported

Response: We have added sample sizes to the figure legends and Method section

Figure 6 seems overly busy and could be divided into two figures. At least e,f, g and h could be better labelled.

Response: As indicated above, we have moved the pie charts and box plots in this figure to a new Supplementary Fig. S9.

Reviewer #2 (Remarks to the Author):

Increasing evidence suggests that the midbrain periaqueductal gray (PAG), especially the dorsal portion in mammals, is a conserved brain center critical for orchestrating the basic action plan for the execution of instinctive survival behaviors that include escape, attack, parenting and breathing. Many studies suggest the PAG is remarkably conserved across phyla and that it also acts as a critical output node for the production of vocal behaviors. Most of what we know about the functional role of PAG derives from work in tetrapods, with most studies performed in mammals, and arguments that PAG acts a conserved structure in the production of innate survival behaviors are based primarily on hodological arguments of how it is connected to pathways necessary for the production of these innate behaviors. Careful physiological functional studies have not been performed in non-tetrapod species.

In a physiological tour-de-force, Schuppe et al. perform a series of careful experiments to show that PAG is both necessary and sufficient for the production of the complex vocalizations that midshipman fish produce. This study therefore provides important new evidence that the PAG, at least in the context of the production of vocal behaviors, acts as a conserved motor output node for the orchestration of these complex innate behaviors. In addition to this important evolutionary contribution, the authors also take full advantage of the behavioral repertoire of these fish to show that the PAG, but not its downstream hindbrain targets, encode two distinct types of vocal behaviors, a behavioral dichotomy that can't be easily exploited in other animal models. Specifically, using fluorescent In Situ hybridization of the immediate early gene c-fos they are able to show, using a well-balanced experimental design coupled with differential cellular localization of nuclear vs. cytoplasmic transcripts, that neurons in portions of the PAG, specifically the caudal superficial portion (PAGcs), show differential activation of subsets of

neurons for these two different vocal behaviors, a pattern that is not observed in the target vocal prepacemaker nucleus (VPP). This finding implies that unique subsets of neurons in this structure are activated to produce these different vocal behaviors. Finally, the authors show convincingly that stimulation of PAGcs, either by direct activation of glutamatergic neurons or by disinhibition of these same areas evokes vocalizations that have many of the characteristics of natural calls with latencies being significantly longer when vocalizations are evoked by disinhibition, results that mirror those obtained in mammals.

Primary comments:

1. The authors show clear differential activation in PAG for the different context-dependent vocalizations. This is implicitly interpreted as being caused by motor-based activation of these different neuronal subsets. However, these findings could potentially also be explained by differential auditory activation given that the PAG is known to be highly integrative. Ideally, it would be good to provide a control where IEG expression is measured in de-vocalized fish (probably the best control). It is possible that fish will not vocalize under such conditions or that such experiments are not feasible. If that is the case, the authors should certainly consider discussing the role that auditory inputs might play in driving *c-fos* expression in these parts of PAG. It is even possible that the authors might well have information (maybe PAG does not receive auditory inputs in midshipman fish?) that address this concern and make it a moot point.

Response: With regards to de-vocalization, one of the authors (A. Bass) has attempted this in the past, however gaining peripheral accessibility to the vocal nerve without incurring significant bleeding is not possible in this species of toadfish. To address the possible effect of audition on the results, we measured *c-fos*+ cell numbers in males from an earlier experiment (not reported in the original submission) that heard courtship hums from another male with its own nest that was separated by a mesh divider in the same tank (see Fig. 3A of Tripp et al., 2020⁷ for an illustration of the tank setup; now cited in manuscript). As in other behavioural experiments, we monitored both nests remotely with hydrophones to ensure that only one male generated courtship hums, while the other remained silent.

Our new analysis demonstrates that silent males in the same tank with a humming male exhibited a greater number of *c-fos*+ cells compared to silent controls, although this was only found for the PAGcs zone. Importantly, expression in both the PAGcs and PAGcd zones was significantly lower than for vocalizing (courtship humming or agonistic calling) males (new Fig. 2i). Thus, elevated *c-fos* activation in the caudal PAG is likely to be strongly dependent on vocalization. We further confirm the latter by adding correlations showing that increased numbers of courtship or agonistic calls is related to increased *c-fos* mRNA expression in the PAG (new Fig. 2j-m). We also note that prior studies of midshipman show that a corollary discharge circuit originating in the PAG's vocal hindbrain target (VPP) to the auditory sensory epithelium of the inner ear would also suppress a fish's own hearing when they were calling (Chagnaud & Bass, 2013⁸; see last sentence in paragraph 3, Results section titled "Differential activation of PAG neurons during vocalization").

2. One of the hallmarks of the PAG, is that it is suggested to contain areas that are differentially activated by different instinctive behaviors. Consistent with this idea, the authors show that different vocal behaviors activate overlapping but nevertheless distinct subsets of neurons in the PAG. The authors also perform an experiment to compare neuronal activation patterns between agonistic vocalizations and a very different behavior, in this case foraging behavior. Here (unless I missed it) they only look at foraging specific *c-fos* expression in PAGcs and PAGcd to compare activation evoked by calling behavior. Consistent with the idea that PAG drives different types of behavior,

only a small subset of neurons is activated by both behaviors. While this is interesting, I would have liked to see analysis of other parts of what is defined as PAG in the fish to show that there is actually an area that is activated primarily by foraging but not vocalization.

Response: We agree that this would be an interesting experiment to do, but it was outside the scope of the main goal of the foraging catFISH studies, which was only to determine whether vocally active neurons in the caudal PAG are also activated when animals perform a non-vocal behaviour, in this case foraging (as noted above, foraging was one of the few behaviours we could reliably evoke without also evoking vocalizations).

The PAG region of midshipman fish can be divided into separate medial and lateral zones (see revised Fig. 2fi), with each likely active across different behavioural states (as shown for mammals; see⁹ and Supplementary Information in Lin et al., 2011⁶). We focused our attention on the lateral PAG since past studies from our lab highlighted this as a potential region important in gating vocalization (e.g., Goodson and Bass, 2002; Kittelberger et al., 2006; Feng and Bass, 2014)¹⁰⁻¹².

3. A large portion of the study is devoted to the use of neurotransmitters or pharmacological agents to initiate fictive vocalizations. While the findings are convincing, it would be useful to provide more information regarding how well injections were restricted to the intended target as well as the inclusion of controls to show that injections of glutamate, say in areas activated by foraging, do not produce vocal behaviors.

Response: In our revised manuscript, we have added analyses and controls for our physiology experiments showing that the effects of glutamate and gabazine on vocal network output (fictive calls) are unlikely due to spread, including (1) experiments that show increasing glutamate volume at the same PAGcs site does not change either the response latency or temporal features of PAGcs-evoked fictive calls (Supplementary Fig. S12), and (2) images of dye injections into the PAGcs and PAGcd showing that dye is localized to that region (Supplementary Fig. S8). Finally, we now better articulate contrasting effects of GLU and gabazine injections in PAGcs and PAGcd on vocal output (see revised figure 7a; see Table S1). Taken together, we believe injections were confined to a particular PAG region.

Minor comments:

Line 42. No references are shown so it is unclear what the authors mean by a circuit of 100s of neurons that can both initiate and patterns complex vocalizations. Do these circuits really only contain 100s of neurons?

Response: We have removed this statement in the revised manuscript.

Line 55 and Figure 1. The statement in the text does not match very well what is represented in the figure given that it does not contain any information about organizational pattern for the limbic stream.

Response: As more clearly stated two sentences later, "Toadfishes, a family of teleost fish (Batrachoididae), have a limbic stream resembling that of mammals, namely direct preoptic-anterior hypothalamic input to a midbrain PAG region that projects, in turn, directly to hindbrain vocal nodes."

Line 64 – please define the term “sensu” which does seem to be defined in the suggested reference.

Response: We have removed this word in the revised manuscript, in part, because we have removed use of the word “motif” or phrase “circuit motif” (e.g., previous title; see above).

Figure 2 – j through o would benefit from a bit more description in the legend regarding the staining pattern shown in the figure.

Response: In the revised figure, we now include high magnification inserts of each caudal PAG zone that highlights the different staining patterns between silent and vocal animals. As noted earlier, we have added higher magnification views of label in PAGcs and PAGcd in bottom row of Fig. 2n-p, and of the PAGrs as inserts to Supplementary Fig. S2a-c.

We also more clearly state in the main text “the striking elevation in *c-fos* label in a lateral periventricular midbrain region...” (Results section titled “Differential activation of PAG neurons during vocalization”).

Figure 3 – it might be good to have a panel showing overlaying the biotin injection site with a *c-fos* staining of the VPP during vocalization.

Response: We have now added a representative image of biotin-labelled neurons following a VPP injection (Fig. 6a and Supplemental Fig. 1c,f).

Line 153 – typo “...calls [had] high amounts ...”

Response: We revised this sentence to fix the typo.

Line 170 – It would be helpful to get a better sense of the actual number of neurons showing *c-fos* staining in the PAGcs and PAGcd for both foraging and vocalizing because the figures only show percentages. Also the figure legend describing l through n is difficult to follow. Please consider revising.

Response: In the revised manuscript, we now add the number of neurons that exhibit *cfos* intron expression (see Fig. 3f).

Line 192 – this is just a suggestion. But it might be nice to introduce this section by stating that behaviors could be initiated either by direct activation of by disinhibition and then perhaps provide examples of how these different strategies are used in other systems or contexts.

Response: We assume the reviewer is referring to the Results section, starting on line 183, that is titled “Glutamatergic vs GABAergic neuron activation in PAG during vocalization”. We believe, especially given space constraints, that this level of broad comparisons across motor systems is best left to a more complete review of this topic. In response to a comment from Reviewer 3 (see 9.7 below), we have now integrated these results in the context of the midshipman PAG having similar vocally active neuron phenotypes as in mammals (end of paragraph 2 in first section of the Results: “We also found greater activation of glutamatergic compared to GABAergic *c-fos*+ cells in both caudal zones during courtship (~75%) and agonistic ~60%) calling Supplementary Fig. S4), which is another PAG character shared with mice.”

It might be worth reminding the reader how to think of the VPP and what would likely be its analogue or homologue in mammals and birds.

Response: Comparing brain nuclei and cell-types between different taxa is a multi-faceted effort, which should include molecular and physiological evidence. Here, we only sought to compare the differential roles of the PAG and VPP in the patterning of context-specific calls in midshipman. Thus, our main point is that vocal pre-motor areas within the hindbrain are largely responsible for sculpting the *basic acoustic unit*, with the PAG being more responsible for patterning the temporal features of complex, social context-specific vocalizations such as a midshipman’s grunt trains/bouts and growls that are an amalgam of the temporal features of grunts and hums.

Line 324 (and in several other places) – Fig 8 is written as Figure 80

Response: There is no longer a Fig. 8, but all such typos have been corrected.

Reviewer: Marc F Schmidt

Reviewer #3 (Remarks to the Author):

In this paper, Schuppe and colleagues analyze activity in the periaqueductal gray (PAG) nucleus of the brains of a type of teleost fish known as toadfish, during two types of vocalization events (courtship humming calling and defensive grunts/growls). The authors use c-fos staining to claim that these different types of vocalization episodes are correlated with different spatial patterns of c-fos activity in the PAG. They then perform tracing studies to claim that the zones they delimited in the PAG have connections with a hindbrain site known as the vocal prepacemaker nucleus (VPP), which previous studies have shown to be part of the vocalization centers in the fish brain. Next, they perform double in situ hybridization with temporal and spatial resolution (catFISH technique) to probe the responsivity of each PAG neuron, concluding that most cells are responsive to either one or the other type of vocalization situation (courtship humming calling or defensive grunts/growls). After conducting staining experiments that claim that most c-fos responsive cells are glutamatergic, the authors conduct a series of pharmacological manipulation experiments with glutamate receptor antagonists or glutamate injections to probe the necessity and sufficiency of the PAG for both types of vocalization outputs.

The experiments in this manuscript lack important controls, were not properly interpreted or are not convincing, and therefore I do not feel that the conclusions drawn from the data are supported.

Major points:

1) Several hypotheses in the paper are either presented as evidence or simply not supported by the data. For example, the observation of c-fos staining in the PAG during episodes of vocalization is certainly much distant from the amount of evidence needed to claim that the nucleus is a “vocalization center”, as seems to be claimed by the authors throughout the study. An example of this is found in line 113: “These results conclusively demonstrate active involvement of lateral PAG neurons in vocal control in toadfishes”; this is an overstatement, as one would not be able to conclude that a nucleus is involved in controlling a certain motor behavior based solely on c-fos staining patterns.

Response: We agree that some results (and the title – see top of this document) were overstated in the previous version of the manuscript. In the revision, we have removed “vocalization center” and made a focused effort to more objectively summarize our results

2) The separation of PAG into PAGr, PAGcd and PAGcs “zones” based on c-fos staining is not convincing. Activity in these areas is not spatially coherent and there is no neuroanatomical indication that each constitutes a functional unit. Other studies even considered PAGcd to be a separate nucleus. Nomenclature issues apart, the c-fos stainings are not convincing in delimiting coherent patterns of activity to justify this zonal categorization, as the somata are diffusively distributed in the multilayered structure that the authors claim to be the PAGcd.

Response: We appreciate the reviewer’s concerns with the zonal categorization. First, we

apologize for our mistake to have suggested that the PAGcd was recognized as a part of another nucleus (lines 109-110); this was not the case.

As further evidence to support our recognition of three PAG zones (one rostral [PAGrs], two caudal[PAGcs/PAGcd]), we now provide more information on the connectivity of each zone based on prior unpublished neuroanatomical tracing studies (end of paragraph 1, first section of the Results that is titled "Differential activation of PAG neurons during vocalization"): "The caudal zones are distinguished, however, by connectivity to the cerebellum (predominantly PAGrs), midbrain tectum (predominantly PAGcs, PAGcd), and midbrain auditory torus (PAGrs, PAGcs) (Fig. S2), and as reported below, patterns of *c-fos* mRNA expression and pharmacological manipulations." The legend for Supplementary Fig. S2 provides more details of the neuroanatomical experiments.

Supplementary Table S1 summarizes the results of pharmacological manipulations (also see responses to points 10 and 11 below).

We also believe that recognizing the PAGcd as part of the PAG is the most parsimonious course of action to follow at this time, given the paucity of functional information for this part of the midbrain in teleost fishes, which made it highly problematic for comparisons to any other previously identified region. We also did not want to invent yet another name for a midbrain region in a teleost that might mean little to colleagues that study tetrapods. We could make some tentative comparisons to zebrafish, but here too the paucity of functional information regarding this region in zebrafish makes the comparisons highly speculative at this time (such speculation is best left for a review).

3) Importantly, Fig. 2i shows that there is no increase in the numbers of c-fos+ cells during courtship as compared to controls (overall statistical significance in the ANOVA test is not enough). Another problem is that the image in Fig. 2n is not representative, as the graph in panel i does not indicate any statistically significant difference compared to the 'silent' controls, but the image shown does display a clear difference.

Response: In the previous draft of the manuscript we reported a significant effect in the rostral, PAGrs zone between silent controls and agonistic, but not courtship, calling males. However, the results were more striking for both caudal PAG zones, and especially the PAGcs and PAGcd, where there were significant increases in *c-fos*+ cell number within both zones. After redoing cell counts using *Cellpose v2*, an automated machine learning cell counting package, we still found significantly elevated *c-fos*+ cell number in both caudal PAG zones of males making either agonistic or courtship calls, and in the PAGcs of agonistic versus courtship humming males. However, the results for the PAGrs were no longer significant. We have revised the style of data presentation in the figure to make these differences more visually clear. We also moved the PAGrs results to a new Supplementary Fig. S3 so that all illustrations of significant results could be enlarged for greater clarity. We have also added more details about cell counts to the Method section (see "Image analysis").

4) It is also hard to judge if the c-fos cell counting was properly done, as the low-mag images in Fig. 2j-o do not make it clear that the green 'dots' dispersed throughout the image are real green-stained c-fos+ cellular nuclei. I see at least 10 green specs that do not coincide with the underlying blue SYTOX nuclear staining.

Response: In the revised manuscript, we have addressed this in two ways that greatly improve the rigor and clarity of our *c-fos* mapping. (1) As noted above, we went back and performed unbiased cell counts using *Cellpose2*, an automated machine learning based cell counting package. We took advantage of their base 'cyto' model and performed additional training on 6 PAGcs images. We found that even with this limited training, our updated 'cyto' model could generalize across brain regions. Importantly, these *Cellpose* cell counts were highly correlated with our original *c-fos* counts. To further validate these *Cellpose* and human observer counts, a

second individual unaware of the experimental conditions that fish were exposed to performed counts on 30 random images across all four conditions. Again, we found that counts were highly consistent across observers and methods. (2) Next, as mentioned above, we added high magnification inserts that display clear labeling differences between vocal and non-vocal animals.

5) The tracing experiments presented in Fig. 3 are hard to interpret. Based on evidence from previous publications that the hindbrain VPP nucleus controls motor activity necessary for vocalization, the authors decided to conduct retrograde tracing to show that the VPP receives inputs from the PAG.

a. Although the staining shown in the PAG is evident, the separation between the staining in the different “zones” defined by the authors (PAGs, PAGcd and PAGcs) is neither evident nor properly quantified. Important controls are also missing, such as showing that the injection site is restricted to the VPP or that other known VPP inputs are also labeled.

Response: Our previous publications already demonstrated that the lateral PAG is connected to VPP following anterograde transport from neurobiotin injections into the PAG (Goodson Bass 2002, Kittelberger et al. 2006)^{10,11}. Those done in Goodson and Bass (2002) were large injections and therefore we could not accurately tell if both caudal zones were connected to VPP. The neuroanatomical findings reported in our new manuscript verify our previously established connection via retrograde transport of neurobiotin from VPP; importantly, this includes showing that the PAGcd identified in the current manuscript is also connected to the vocal hindbrain circuit. In the revision, we provide more clear images that show filled cells in PAGcs and PAGcd (see Supplementary Fig. S1d).

In the revised manuscript, we also add a representative images that shows labeled VPP neurons following a neurobiotin injection into VPP (Supplementary Fig. S1c,e; and Fig. 6a). This injection also confirmed VPP’s previously reported extensive connectivity to the pacemaker-motoneuron (VPN-VMN) part of the hindbrain circuitry. Finally, in Fig. 6b we map non-vocal sites (n=2) that missed VPP.

b. Importantly, detecting retrograde labeling from the VPP in the PAG is not enough to classify the latter as part of a vocal circuit or as a “vocal hotspot”, as claimed by the authors in line 126 (moreover, the word “hotspot” is most certainly meaningless jargon in this sentence).

Response: We agree and have removed this language and provided more clear interpretation of these findings.

6) The heterogeneity of cellular responses (as judged by c-fos expression) during different types of vocalization events was investigated using the catFISH procedure (Fig. 4). This experiment would be interesting to define the map of activity concomitant with (not “in response to”, as argued by the authors) vocalization events, but the results are far from convincing:

6.1 - The critical controls in Suppl. Fig. 1 are simply not good enough: (a) I see clear green staining in Suppl. Fig. 1b (Silent-Agonistic situation), in which case one would expect to see only red staining related to the c-fos intronic probe; (b) the red staining related to the c-fos intronic probe is not evident anywhere in Suppl. Fig. 1 or in Fig. 4; there should be very clear foci (preferably two per nucleus) inside the blue nuclei in the picture, and I do not see any; the images in Suppl. Fig. 2e,g are better but still not clear enough since all fluorescent channels are merged; (c) likewise, the green staining in

Suppl. Fig. 1c (Agonistic-Silent situation) should be predominant, but I cannot judge to which extent red staining is also present, since only the merged image is presented and some red staining is clearly apparent in some cells; importantly, the green staining should be both nuclear and cytoplasmic, as it is the result of staining with a *c-fos* probe designed against the coding region, but I cannot judge if the blurry (and sometimes punctate) pattern of green staining is truly cytoplasmic in these low-mag images; (d) finally, no quantification is presented for these critical control experiments in Suppl. Fig. 1.

Response:

(6.1 a) We have now clarified what the catFISH staining should look like by including a new schematic diagram (new Fig. 3e). As described in Ishii and colleagues (2017⁵), neurons active only in behavior 2 (most recent behaviour) should exhibit yellow nuclear staining, which is a product of the overlap between the two probes. We also address this by separating the channels on several locations where intronic signal is present in the PAGcs (see Figure 3 legend). We have not added this for the panels illustrating PAGcd label (Fig. 3h, j, l, n) to avoid the figure being too crowded; however, we can add this if the reviewer considers it necessary.

(6.1 b-c) For figure 3, we also now add high magnification images of both PAGcs and PAGcd. In these images, there is clear yellow staining (overlap between *c-fos* coding and *c-fos* intron, Fig. 3n,o). We also made the recommended change of including unmerged example regions, but we are concerned about the figure becoming too crowded. In the previous version, we were following the seemingly accepted convention of previous catFISH figures (see Lin et al., 2010, Ishii et al., 2017)^{5,6}.

Importantly, we draw attention to the fact that our images look highly similar to that of Ishii and colleagues (2017⁵; their Fig. 7) and Lin and colleagues (2011⁶; their Fig. 1). We also followed the exact staining protocol as Ishii and colleagues (2017⁵), including using the same reagents to amplify the signals. We took these steps to ensure that we could reproduce catFISH staining in a non-mammal. As in these papers, an observer evaluated expression in a z-stacks with 2 μ m steps. This allowed us to view the expression of staining in individual cells. We have added these details to the “Image analysis” section of the methods

(6.1 d) In the revision, we now include a quantification of both the total number of cells with intronic expression (those associated with behaviors produced in the last 5 minutes before sacrificing the animal) and the total number of cells with cytoplasmic expression for the timecourse controls (new Supplementary Fig. S5c-f). This analysis is comparable to that initially performed by Lin and colleagues (2011)⁶. Specifically, we see that there is a significantly greater number of *c-fos* intronic signal in animals that vocalized in the 5 minutes before sacrifice (behavioural period 2). Conversely, agonistic-silent animals had significantly high cytoplasmic expression (reflective of neuronal activation during behavioural period 1 – agonistic calling). Finally, we also show that levels of *c-fos* intronic expression in the PAGcs are positively correlated with total agonistic behavior.

6.2 – Most significantly, the same criticism goes for the whole of Fig. 4: no evident red staining is seen anywhere in these low-mag merged images, and the green staining is either punctate (and apparently not coinciding with any blue nucleus) or blurry and not convincing as cytoplasmic staining.

Response: We revised this figure to now include higher magnification confocal max intensity projections. Also, as mentioned above, our staining is consistent with previous papers. High magnification maximum intensity projections in Fig. 3 (ni-nii,oi-iii) show clear cytoplasmic and nuclear (intron) staining. As mentioned above, we also include an insert in Fig 3 (h, j, l) with the

channels unmerged. These clearly show green staining that surrounds blue nuclei. Importantly, all catFISH images were analyzed by scanning through z-stacks. This allowed to accurately assign nuclear and cytoplasmic signal to individual nuclei. We have now added these details to the Methods section of the paper.

6.3 - Moreover, the graphs in Fig. 4f claim that consecutive episodes of vocalization of the same type (courtship-courtship, for example) lead to largely overlapping staining with the c-fos coding and intronic probes, as should be the case if the catFISH procedure worked correctly, but I cannot see such significant level of overlap in the corresponding image in panel g. Likewise, the agonistic-agonistic situation should result in largely overlapping subpopulations of c-fos+ cells according to the graph in panel f, but I cannot see that in the image in panel h. Without such controls, the interpretation of panel i, which contains the critical comparison between the “agonistic” and the “courtship” situations, is rendered impossible.

Response: The extent of overlap between behaviour 1 and behaviour 2 that we find (~65-90%) is highly consistent with previous studies that have used this technique. For instance, in Lin et al. (2011⁶), mice that mated twice had between ~40-90% overlap. Notably, their findings demonstrated reactivation across brain areas analyzed (Lin et al., 2011)⁶. Furthermore, mice that performed the same behaviour twice in Ishii and colleagues (2017)⁵ had similar levels of reactivation. In this way, we think that our results from animals that performed the same behaviour twice is highly consistent with all of the currently available studies that have used this technique. Our analysis in Fig. 3 shows silent-silent animals still have occasional intronic signal; this is likely expected of a brain area like the PAG that is involved in diverse behaviours (our revised figure 3 shows a single neuron with nuclear intron expression; see insert in ‘h’). Regardless, animals in our study that performed different vocal behaviours had significant neuronal reactivation (PAGcs: ~60%; PAGcd:~40%; see Fig. 3). Thus, there is a consistent signal that we can detect.

6.4 - I also think it is very confusing to represent the bar graphs in Fig. 4 using ‘green’, ‘red’ and ‘yellow’ colors, as these colors might give the impression that the ‘yellow’ bars refer to concomitant staining with both c-fos coding (green) + c-fos intronic (red) probes, which is certainly not the case.

Response: We have altered the color scheme of the figure to avoid this confusion and streamline the interpretation of the results presented in the figure.

6.5 - Finally, the numbers of replicates in each comparison are not properly described in the figure legend (does n=5 mean 5 fishes or 5 sections from a smaller number of fish?).

Response: We used 5 individual male fish per condition. For each fish, we analyzed expression in 2-3 consecutive sections that included both PAGcs and PAGcd. We now include these details in the figure legend and Methods section.

7) Even considering that the analyses and quantification in this catFISH experiment are sound, the conclusions that there is little overlap between the c-fos staining for the agonistic versus the courtship vocalizations and that such small overlap implies that both situations activate largely distinct subpopulations of PAG cells are not supported by the data. The graph in Fig. 4f shows close to 50% overlap in both PAG regions analyzed, so the conclusion that the subpopulations of active PAG cells are disjunct is not warranted. And in the PAGcs, which later experiments in the paper interpreted to be the primary output to the vocal hindbrain centers to initiate vocalization, the level of overlap of c-fos staining for both types of vocalization events is close to 70%.

Response: We agree that a more nuanced interpretation of these findings is needed. Our findings suggest that there is significantly less reactivation of PAGcs and PAGcd when animals perform two different types of vocalizations (courtship-agonistic) compared to when they perform the same behaviours twice (courtship-courtship, agonistic-agonistic). Consistent with interpretations in previous catFISH studies, we interpret this to mean that there are some neurons associated with agonistic calling that were not activated during courtship calling (note that we do not state “that both situations [courtship calling and agonistic calling) activate largely distinct subpopulations of PAG cells”; we are conservative in only claiming this for agonistic calling as these fish will not courtship call after a first behavioural period of agonistic calling). Nonetheless, we agree that there is substantial overlap between the two populations. This implies that there are some neurons that are likely activated in both agonistic and courtship calling contexts. We have modified the Results and their Discussion to more accurately reflect this interpretation that PAGc may exhibit some multi-functionality associated with vocal-motor behaviours (see sentences 6 and 7 in paragraph 2 of the Results section titled “Call specific activation in PAG, but not hindbrain” and sentence 3 in paragraph 2 in the Discussion).

8) Irrespective of these technical limitations and the poor quality of staining, microscopy images, and quantification, the authors conclude that there is little overlap between the c-fos staining in PAG neurons during vocalization and during a non-vocalizing activity (foraging) in Fig. 4m,n and that this result implies there is “little evidence of functionality of the PAG zones outside of their role in calling” (line 179). I cannot objectively understand this statement and I do not see how the analysis of c-fos activity in two brain regions when the animal is performing only two types of tasks can lead to any conclusion about functionality, of either the circuit or the specific nuclei analyzed.

Response: We agree that we originally overstated some claims regarding the *c-fos* staining. To address these concerns, we remove statements about functionality and *c-fos* staining. Instead, we now state that the *c-fos* staining is associated with a particular behaviour.

9) The characterization of c-Fos+ cells in the PAG relative to their glutamatergic or GABAergic phenotypes is based on unconvincing staining as well (Fig. 5):

9.1 and 9.2 - The staining in Fig. 5d-j used antibodies against glutamate and GABA, which is far from ideal, as the resulting fluorescent patterns are diffuse, difficult to correlate with specific cell bodies and with high non-specific background staining. I see very clear overlapping staining for glutamate and GABA in a number of instances, such as in the insets in Fig. 5e,f and in Suppl. Fig. 3, which is hard to understand and reconcile with the conclusions that the vocalization-related c-fos activity is found mostly in glutamatergic cells.

Response: In the original version of the manuscript, we presented a maximum intensity projection of this cell dense region. While it seems like there was overlap, analysis of z-stacks illustrates that there is no overlap in the signal. Importantly, analysis was done on z-stacks and not maximum intensity projections. This allowed us to clearly distinguish the signals. We now present a z-stack montage in Supplementary Fig. S4.

Furthermore, we add more details about the previous validations of the GLU and GABA antibodies to the Methods. Specifically, we now state: “Previous experiments have extensively validated both antibodies. For instance, the GLU antibody specifically labels neurons expressing a fluorescent protein under control of the VGLUT2 promoter¹³, and preadsorption with glutamate eliminates labeling in another teleost¹⁴. Likewise, with the GABA antibody, elimination of label following preadsorption of the antibody with GABA has been reported in a broad range of species^{15–17}.”

9.3 - Moreover, the c-fos staining in Fig. 5 is not good, as it is a mixture of punctate

staining, background staining, and some staining that appears to occupy the whole cell body. Again, it is hard to correlate the staining with the underlying nuclear staining, as all images are low-mag, presented as merged fluorescent images, and no SYTOX staining was performed or shown.

Response: We appreciate the critical interpretation of figures and the need to improve them and our interpretation of the data. However, we respectfully disagree about the staining being background staining. We only see *c-fos* expression where cells are present and not in parts of the tissue where there is no background. Importantly, scanning z-stacks show that *c-fos* staining overlaps with a neuron's cytoplasm.

9.4 - Importantly, the method the authors used for quantifying the levels of staining for either glutamate or GABA in Fig. 5a,b is not ideal: numbers of c-fos+ cells that co-stain for GLU or GABA do not mean much, as the authors themselves state that the overall numbers of "cells" stained with GLU or GABA differ dramatically in all PAG zones. Therefore, the best strategy would be to quantify the percentages of c-Fos+ cells that are either GLU or GABA-positive, a procedure that seems to have been performed and shown in Fig. 5d, in pie graphs that are neither quantitative, nor statistically explored, nor properly explained (scale?).

Response: In the revision, we have pared down these analyses as suggested by the reviewer. Specifically, we only quantify percentages for each zone. We have also modified the figure to make all the pie charts the same size (Supplementary Fig. S4b).

9.5 - These experiments must preferably be conducted with antibodies against glutamate or GABA vesicle markers (vGLUT, vGAT, or GAD) or preferably still with in situ hybridization probes against the respective genes, as these staining patterns would be very clearly associated with cell bodies, allowing the precise quantification of an overlap with the c-Fos staining pattern.

Response: While we appreciate that it is becoming more common to use in situ hybridization to phenotype neurons, this is not trivial to do in non-model systems, especially teleost fish. Unlike birds and mammals that have well annotated genomes, fish genomes are less common. Currently, there is no midshipman genome, and thus purchase of short HCR probes is not possible. Instead, we have cloned, sequenced, and generated probes (as was common before HCR probes). Importantly, the GLU and GABA antibodies that we use, which have been used in studies of other well-established fish (lamprey and zebrafish¹⁸⁻²⁰) and mammalian²⁰⁻²² study species to phenotype GABA+ or GLU+ neurons (also see paragraph 2 above in response to 9.1 and 9.2). This technique is well established and is currently still being used in recent *Nature Communication* papers (see references²⁰⁻²² below). As mentioned above, we now provide information on the specificity of these antibodies. For example, other studies have shown overlap between GLU antibody and vesicular glutamate (VGLUT) antibody staining²³ and with VGLUT2 specific lines¹³.

We believe that the additional of all of these methodological details (see point 9.1 for more details) are sufficient to justify the use of the GLU and GABA antibody. To further satisfy the reviewer's concerns we provide some evidence VGLUT2 mRNA and GLU antibody staining is highly similar (see below). This VGLUT2 mRNA probe part of a larger study, involving many collaborators, and we prefer to not compromise findings of that ongoing study by switching to using VGLUT. However, we have provided an image here to further validate the antibody for the reviewer.

9.6 – The sentences in line 195 discuss that the VPP receives inputs from the PAG and that glutamatergic signaling in the VPP is important for vocalization, but no experiments were performed to show that glutamatergic projections from the PAG vocalization responsive cells reach the VPP.

Response: We agree that “no experiments were performed to show that glutamatergic projections from the PAG vocalization responsive cells reach the VPP” and have deleted the paragraph beginning on line 195. We do, however, believe that the GLU injections that we performed into VPP together with the super resolution images that illustrate GLU contacts onto VPP somata (Fig. 6c), show that “glutamatergic signaling in the VPP is important for vocalization”.

9.7 - In sum, I am not convinced, based on the data and graphs presented, that “activation of GLU-, but not GABA, containing PAGc neurons are essential for initiating social context-specific vocal output”, as the authors claim in line 196. Even if the calculations were correct, why would the quantification of c-fos/GLU/GABA co-staining indicate anything about the essentiality of a nucleus in initiating a behavioral output?

Response: We agree with this criticism. As stated above, we have eliminated the original

paragraph beginning on line 195. To address this comment, we have now simplified our interpretation of these results and simply state (section 1 of the Results that is titled “Differential activation of PAG neurons during vocalization”): “We also found greater activation of glutamatergic compared to GABAergic *c-fos+* cells in both caudal zones during courtship (~75%) and agonistic (~60%) calling Supplementary Fig. S4), which is another PAG character shared with mice.”

10) In the experiments to pharmacologically manipulate the activity of PAG neurons (with either GABA agonist or glutamate receptor antagonists) in Fig. 6, because the animals are anesthetized the authors used a GABA receptor antagonist injection to elicit patterns of vocalization measurable by evaluating the occipital nerve, instead of using the more natural situation in which the animals vocalize either to the dummy fish or to attract females in the basal humming vocal events. Even though some effort is undertaken to compare the features of pharmacologically-evoked calls and the natural responses using dimensionality reduction algorithms (UMAP) in Fig. 6, the results do not seem to show that these responses are similar, as claimed by the authors: for example, I can clearly see a difference between the responses of the natural grunts (in green in Fig. 6f) and the grunts induced by injection of glutamate into the PAGcs (in red).

Response: In the revision, we have added additional statistical analyses and better interpret these findings. We now compare the duration and variability in PRR and amplitude of fictive calls to natural calls (see Fig. 4, Supplementary Table S1).

It was never our intention, for example, to imply that GLU-evoked fictive calls were identical to natural calls. Similarly, both chemo- and ontogenetically elicited calls in mice differ in several notable acoustic features (Tschida et al., 2019)². Nonetheless, we find it extremely exciting that stimulation of PAG neurons in a non-mammal is able to generate both agonistic and courtship calls that exhibit complex changes in PRR and amplitude that resemble natural calls.

11) Moreover, the authors do not show the images from control experiments to evaluate that the glutamate injections into the PAG target solely this nucleus and, importantly, just one of the PAG zones analyzed. Since the PAGcd and PAGcs responses are being contrasted, these controls are critical. Given the leakiness and potential variability associated with stereotaxic pharmacological injections into the brains of a non-standard species, I consider that these assays are prone to the vagaries of experimentation and I am not sure that the data shown are enough to assess such variability in order to allow meaningful conclusions to be drawn about the sufficiency of the PAG and its “zones” in eliciting various types of vocalization patterns, as claimed by the authors.

Response: In the revised manuscript, we have addressed this comment in two ways. First, we add new images to Supplementary Fig. S8 that show the spread of dye in these injections appears localized to the PAG zone that is being manipulated. Second, we provide new control experiments that illustrate that increasing the amount of GLU does not change call features (Supplementary Fig. S12).

12) All supporting data must be shown. ‘Data not shown’ statements are not sufficient and are not allowed as per Nature’s reporting guidelines.

Response: We provide this data in a new Supplementary Fig. S12.

13) Given these technical and interpretational problems, I am not convinced by the pharmacological manipulation experiments and therefore cannot support a publication stating that the PAG is both necessary and sufficient to drive different types of vocalization patterns in the model fish used (line 256).

Response: We agree that this was too strong of a statement and have removed this language in the revised version of the manuscript. Our conclusions are now more tempered and consistent with the results presented. For example, see beginning of the last paragraph in the Results section titled “Blocking GABAergic action in PAG only generates agonistic-like vocal output”: “ In sum, GLU or gabazine actions in the caudolateral PAG of midshipman fish suggested that separate zones may differentially bias production of different acoustic features and, in turn, call type.”

Minor points:

1) There is no need to keep Figure 3 as separate, given the minimally important set of data presented therein for the whole story. Either join these data to another figure or transfer them to the supplementary material.

Response: The original Figure 3 has been deleted, although 2d has been integrated into a new figure (Supplementary Fig. S1).

2) Fig. 1 and the first panel in Fig. 2 should not be presented before the main results. They are prior data and could be transferred either to the supplementary material or simply explained in the text.

Response: Figure 1 is an introductory figure that places the current study of sonic fish into the larger context of the evolution of vertebrate sound production. Given that many scientists and non-scientists alike often do not realize that fish even make communication sounds, we believe this overview is an important and essential comparative framework for our study.

While the first panels of Figure 2 (a-c) include previously described calls, these are all calls recorded during the current experiments (using the set-ups shown in Fig. 2g and h). Additionally, these are the behaviours at the center of the story presented in this manuscript and thus seem essential to its content. We believe these are useful for the reader to appreciate the acoustic differences in each call type.

3) The “2” bottom-left inset in Fig. 4n does not correspond to the indicated region in the low-mag image.

Response: We changed the layout of this figure in the revised manuscript and corrected any such errors. In the new figure, we ensured that all high mag inserts correspond to the region demarcated in the low mag image.

4) The meaning of letters ‘a’, ‘b’, etc. above bars in bar graphs after ANOVA should be described in the legend, as being the result of post-hoc pairwise comparisons.

Response: We have added the suggested change the figure legend.

5) Fig. 1 legend is cut short.

Response: We have modified the figure legend in the revised manuscript.

6) Panels in Fig. 2j-o are misaligned.

Response: We changed this layout of this figure in the revised manuscript.

7) The Methods section should be carefully revised, as the text is very confusing and inaccurate in a lot of instances. For example, in the sentences starting in line 504, the authors refer to a “biotin blocking reagent” (without description) and later they refer to a “streptavidin blocking reagent” and to a “blocking buffer”. Each solution in the experiments must be clearly specified (concentrations and reagent manufacturers) and

the methods should be described using a nomenclature system that is both unique and standard.

Response: We have revised the Methods section to add more details, including the reagents in each buffer and the manufacturer information.

8) In line 522, the description of catFISH is confusing. If FITC and DIG labeled probes were used, why have the authors labeled the FITC probe with Cy3? If this is not the case, then the text is confusing and must be rewritten. I imagine that they used an HRP-conjugated anti-DIG antibody, followed by Cy3-tyramide labeling, for the c-fos intron probe (red fluorescence in the microscopy images), but this is not what the text describes. Moreover, the brands and part numbers of all critical reagents must be listed throughout the main text.

Response: We appreciate the reviewer for catching this error in our method section. DIG probes were amplified with cy3-tyramide. We have carefully revised the method sections for accuracy and added the important content pointed out by the reviewer.

Supplemental Materials References

1. Jarvis, E. D., Scharff, C., Grossman, M. R., Ramos, J. A. & Nottebohm, F. For whom the bird sings: Context-dependent gene expression. *Neuron* **21**, 775–788 (1998).
2. Tschida, K. *et al.* A Specialized Neural Circuit Gates Social Vocalizations in the Mouse. *Neuron* **103**, 459–472.e4 (2019).
3. DeMartini, E. E. Spawning success of the male plainfin midshipman. I. Influences of male body size and area of spawning site. *J Exp Mar Biol Ecol* (1988) doi:10.1016/0022-0981(88)90254-7.
4. Sisneros, J. A., Alderks, P. W., Leon, K. & Sniffen, B. Morphometric changes associated with the reproductive cycle and behaviour of the intertidal-nesting, male plainfin midshipman *Porichthys notatus*. *J Fish Biol* (2009) doi:10.1111/j.1095-8649.2008.02104.x.
5. Ishii, K. K. *et al.* A Labeled-Line Neural Circuit for Pheromone-Mediated Sexual Behaviors in Mice. *Neuron* **95**, 123–137.e8 (2017).
6. Lin, D. *et al.* Functional identification of an aggression locus in the mouse hypothalamus. *Nature* **470**, 221–227 (2011).
7. Tripp, J. A., Salas-Allende, I., Makowski, A. & Bass, A. H. Mating behavioral function of preoptic galanin neurons is shared between fish with alternative male reproductive tactics and tetrapods. *Journal of Neuroscience* **40**, 1549–1559 (2020).
8. Chagnaud, B. P. & Bass, A. H. Vocal corollary discharge communicates call duration to vertebrate auditory system. *Journal of Neuroscience* **33**, 18775–18780 (2013).
9. Vaughn, E., Eichhorn, S., Jung, W., Zhuang, X. & Dulac, C. Three-dimensional Interrogation of Cell Types and Instinctive Behavior in the Periaqueductal Gray. *bioRxiv* 2022.06.27.497769 (2022) doi:10.1101/2022.06.27.497769.
10. Goodson, J. L. & Bass, A. H. Vocal-acoustic circuitry and descending vocal pathways in teleost fish: Convergence with terrestrial vertebrates reveals conserved traits. *Journal of Comparative Neurology* **448**, 298–322 (2002).
11. Matthew Kittelberger, J., Land, B. R. & Bass, A. H. Midbrain periaqueductal gray and vocal patterning in a teleost fish. *J Neurophysiol* **96**, 71–85 (2006).
12. Feng, N. Y., Marchaterre, M. A. & Bass, A. H. Melatonin receptor expression in vocal, auditory, and neuroendocrine centers of a highly vocal fish, the plainfin midshipman (*Porichthys notatus*). *Journal of Comparative Neurology* **527**, 1362–1377 (2019).
13. Pedroni, A. & Ampatzis, K. Large-Scale Analysis of the Diversity and Complexity of the Adult Spinal Cord Neurotransmitter Typology. *iScience* (2019) doi:10.1016/j.isci.2019.09.010.
14. Berg, E. M., Bertuzzi, M. & Ampatzis, K. Complementary expression of calcium binding proteins delineates the functional organization of the locomotor network. *Brain Struct Funct* (2018) doi:10.1007/s00429-018-1622-4.
15. Webber, M. P., Thomson, J. W. S., Buckland-Nicks, J., Croll, R. P. & Wyeth, R. C. GABA-, histamine-, and FMRFamide-immunoreactivity in the visual, vestibular and central nervous systems of *Hermisenda crassicornis*. *Journal of Comparative Neurology* (2017) doi:10.1002/cne.24286.

16. Iwano, M. *et al.* Neurons associated with the flip-flop activity in the lateral accessory lobe and ventral protocerebrum of the silkworm moth brain. *Journal of Comparative Neurology* (2010) doi:10.1002/cne.22224.
17. Tripp, J. A. & Bass, A. H. Galanin immunoreactivity is sexually polymorphic in neuroendocrine and vocal-acoustic systems in a teleost fish. *Journal of Comparative Neurology* **528**, 433–452 (2020).
18. Suryanarayana, S. M., Pérez-Fernández, J., Robertson, B. & Grillner, S. Olfaction in Lamprey Pallium Revisited—Dual Projections of Mitral and Tufted Cells. *Cell Rep* (2021) doi:10.1016/j.celrep.2020.108596.
19. Suryanarayana, S. M., Robertson, B., Wallén, P. & Grillner, S. The Lamprey Pallium Provides a Blueprint of the Mammalian Layered Cortex. *Current Biology* (2017) doi:10.1016/j.cub.2017.09.034.
20. Huang, C. X. *et al.* An injury-induced serotonergic neuron subpopulation contributes to axon regrowth and function restoration after spinal cord injury in zebrafish. *Nat Commun* (2021) doi:10.1038/s41467-021-27419-w.
21. Feng, H. *et al.* Orexin signaling modulates synchronized excitation in the sublateralodorsal tegmental nucleus to stabilize REM sleep. *Nat Commun* (2020) doi:10.1038/s41467-020-17401-3.
22. Guo, W. *et al.* A Brainstem reticulotegmental neural ensemble drives acoustic startle reflexes. *Nat Commun* (2021) doi:10.1038/s41467-021-26723-9.
23. Flaive, A., Cabelguen, J. M. & Ryczko, D. The serotonin reuptake blocker citalopram destabilizes fictive locomotor activity in salamander axial circuits through 5-HT1A receptors. *J Neurophysiol* (2020) doi:10.1152/jn.00179.2020.

REVIEWERS' COMMENTS

Reviewer #1 (Remarks to the Author):

I appreciate the authors response to my comments. The additional details have strengthened the manuscripts.

Reviewer #2 (Remarks to the Author):

The authors have done a thorough job revising their manuscript. I find it a much stronger study even though I found the original submission quite compelling. They have addressed all of my concerns to my satisfaction. The other two reviewers (especially reviewer #3) raised many valid point as well and I am pleased that they modified many aspects in their study top dress these important comments. I only have a few minor suggestions.

Line 176 – I find the statement “(nesting males gain weight and feed)” a little awkward. I would recommend providing a brief description here (or in the supplemental) of the type of motor engagement that is involved in foraging to get a better sense of the types of movements that are performed.

Line 183 – Figure 3s seems like an important control to show that neurons activated by foraging are not re-activated by agonistic calling. I feel like a half sentence to make its importance more clear would be useful especially since 3r and 3s look so similar and it is easy to miss the label on the y-axis.

Line 203-204. This section has the potential of confusing the reader, I think. I certainly got confused by the simple statement about how fictive calling is generated (I assume by injection of gabazine in the limbic forebrain; in POA?; even description in lines 590-91 seems a bit vague). These experiments are then coupled with additional injections in PAG. These are important experiments and I feel like a bit more detail (in the form of a figure showing double injections sites; in the main figure) would be helpful even if it might be obvious to some that these types of manipulations have been performed in the past to generate fictive calling.

Reviewer #3 (Remarks to the Author):

In the revised form of this paper, Schuppe and colleagues analyze activity in the periaqueductal gray (PAG) nucleus of toadfish brains, during two types of vocalization events (courtship humming calling and defensive grunts/growls). The authors use c-fos staining to show that these different types of vocalization episodes are correlated with different spatial patterns of c-fos activity in the PAG. They also perform double in situ hybridization with temporal and spatial resolution (catFISH technique) to probe the responsivity of each PAG neuron, concluding that individual neurons are active during both courtship and agonistic or only agonistic calling, with few co-activated during a non-vocal behavior. Lastly, the authors conduct pharmacological manipulation experiments with glutamate receptor antagonists or glutamate injections to probe the participation of the PAG in eliciting vocalization outputs.

I consider this revised version of the manuscript much improved. The authors have made great efforts to include new analyses and changes to the text and figures in response to most of my previous criticisms. When they have not done so, the authors convinced me that these analyses were either not necessary or that their hypotheses relied on previously published data.

With further explanations about statistical significance, sample sizes and effect sizes, I am convinced that the main claims of the paper are supported by the additional experiments. Although

more could be done to parse out the contribution of glutamatergic inputs in eliciting the different types of vocalization patterns explored in the study, what is presented is convincing and therefore suggestions of additional experiments are not appropriate at this stage.

I would like to further note that the authors' responses to the criticisms by the other two reviewers strengthened by belief the data and conclusions are sound.

A few extra minor suggestions that the authors could pay attention to during copy-editing (all text-related):

l. 48 – corticobulbar pathway (singular form)

l. 64 – space after “documented”

l. 71 – “100s of msec” might be better written as “hundreds of milliseconds”. If it is deemed necessary to refer to ‘milliseconds’ in its symbol form, the correct is ‘ms’ as per Nature’s guidelines. Same suggestion goes for ‘sec’, which should be replaced by either ‘seconds’ or its correct symbol ‘s’.

l. 128 – mis-positioned reference 8.

l. 335 – the three dots in the quoted sentence should be placed between parentheses.

REVIEWER COMMENTS

Reviewer #1 (Remarks to the Author):

I appreciate the authors response to my comments. The additional details have strengthened the manuscripts.

Reviewer #2 (Remarks to the Author):

The authors have done a thorough job revising their manuscript. I find it a much stronger study even though I found the original submission quite compelling. They have addressed all of my concerns to my satisfaction. The other two reviewers (especially reviewer #3) raised many valid point as well and I am pleased that they modified many aspects in their study to address these important comments. I only have a few minor suggestions.

Line 176 – I find the statement “(nesting males gain weight and feed)” a little awkward. I would recommend providing a brief description here (or in the supplemental) of the type of motor engagement that is involved in foraging to get a better sense of the types of movements that are performed.

Response: We agree that this could be clearer. In the revision, we have modified this sentence to now include citations (see sentence 1, paragraph 3 in section titled “Call specific activation in PAG, but not vocal hindbrain”). As suggested by the reviewer we now add more information about the foraging assay to the Methods (see section titled “Foraging” under “Behaviour experiments”). In general, fish remained in the nest and only swam toward the goldfish when it came near the nest. Finally, we include a new video (Video S3) of a male in the nest that had just captured a goldfish. Importantly, this demonstrates like the other conditions fish that fed remained in the nest.

Line 183 – Figure 3s seems like an important control to show that neurons activated by foraging are not re-activated by agonistic calling. I feel like a half sentence to make its importance more clear would be useful especially since 3r and 3s look so similar and it is easy to miss the label on the y-axis.

Response: In the revision, we have modified the text to more clearly articulate that there is greater neural activation (*c-fos* intron expression) in PAGcs and PAGcd (see paragraph 3 in section titled “Call specific activation in PAG, but not vocal hindbrain”). We also stress that the limited neural reactivation in the feeding-agonistic catFISH condition demonstrate that vocal and foraging neurons in the caudolateral PAG are likely distinct populations. As we write, “Compared to males that foraged twice, *c-fos* intronic expression was significantly higher by 3.5 (PAGcs) to 10 (PAGcd) fold in males that made agonistic calls during period 2 (Fig. 3r). Thus, agonistic vocalization was associated with greater neural activity in the caudal PAG compared to foraging. About 75% of these PAGcs neurons, but <5% of PAGcd neurons, were reactivated when animals foraged twice (Fig. 3s). This suggests that caudal PAG neurons involved in vocalization and foraging are largely independent populations.”

Line 203-204. This section has the potential of confusing the reader, I think. I certainly got confused by the simple statement about how fictive calling is generated (I assume by injection of gabazine in the limbic forebrain; in POA?; even description in lines 590-91 seems a bit vague). These experiments are then coupled with additional injections in PAG. These are important experiments and I feel like a bit more detail (in the form of a figure showing double injections sites; in the main figure) would be helpful even if it might be obvious to some that these types of manipulations have been performed in the past to generate fictive calling.

Response: In the revision, we have made several modifications to supplementary Figure S7 to address the reviewer's comments. First, we add a sagittal schematic illustration that shows the first injection of gabazine in the POA to initiate fictive calling and later bilateral injections of GluR blockers (APV/NBQX) into the PAG to silence fictive calling (Fig. S7c). Second, we add an additional image that shows dye that was mixed GluR to mark PAG sites (Fig. S7d).

Reviewer #3 (Remarks to the Author):

In the revised form of this paper, Schuppe and colleagues analyze activity in the periaqueductal gray (PAG) nucleus of toadfish brains, during two types of vocalization events (courtship humming calling and defensive grunts/growls). The authors use c-fos staining to show that these different types of vocalization episodes are correlated with different spatial patterns of c-fos activity in the PAG. They also perform double in situ hybridization with temporal and spatial resolution (catFISH technique) to probe the responsivity of each PAG neuron, concluding that individual neurons are active during both courtship and agonistic or only agonistic calling, with few co-activated during a non-vocal behavior. Lastly, the authors conduct pharmacological manipulation experiments with glutamate receptor antagonists or glutamate injections to probe the participation of the PAG in eliciting vocalization outputs.

I consider this revised version of the manuscript much improved. The authors have made great efforts to include new analyses and changes to the text and figures in response to most of my previous criticisms. When they have not done so, the authors convinced me that these analyses were either not necessary or that their hypotheses relied on previously published data.

With further explanations about statistical significance, sample sizes and effect sizes, I am convinced that the main claims of the paper are supported by the additional experiments. Although more could be done to parse out the contribution of glutamatergic inputs in eliciting the different types of vocalization patterns explored in the study, what is presented is convincing and therefore suggestions of additional experiments are not appropriate at this stage.

I would like to further note that the authors' responses to the criticisms by the other two reviewers strengthened by belief the data and conclusions are sound.

A few extra minor suggestions that the authors could pay attention to during copy-editing (all

text-related):

l. 48 – corticobulbar pathway (singular form)

We fixed this typo.

l. 64 – space after “documented”

We added a space.

l. 71 – “100s of msec” might be better written as “hundreds of milliseconds”. If it is deemed necessary to refer to ‘milliseconds’ in its symbol form, the correct is ‘ms’ as per Nature’s guidelines. Same suggestion goes for ‘sec’, which should be replaced by either ‘seconds’ or its correct symbol ‘s’.

We have made these changes throughout.

l. 128 – mis-positioned reference 8.

Fixed.

l. 335 – the three dots in the quoted sentence should be placed between parentheses.

We have revised this sentence in the revision.